

# The phylogenetic affinities of the bizarre Late Cretaceous Romanian theropod *Balaur bondoc* (Dinosauria, Maniraptora): dromaeosaurid or flightless bird?

Andrea Cau[1], Tom Brougham[2,3] and Darren Naish[2,3]

[1] Earth, Life and Environmental Sciences Department, Alma Mater Studiorum, Bologna University, Italy
[2] Ocean and Earth Science, University of Southampton, Southampton, UK
[3] These authors contributed equally to this work.

Corresponding author
Andrea Cau, cauand@gmail.com

## ABSTRACT

The exceptionally well-preserved Romanian dinosaur *Balaur bondoc* is the most complete theropod known to date from the Upper Cretaceous of Europe. Previous studies of this remarkable taxon have included its phylogenetic interpretation as an aberrant dromaeosaurid with velociraptorine affinities. However, *Balaur* displays a combination of both apparently plesiomorphic and derived bird-like characters. Here, we analyse those features in a phylogenetic revision and show how they challenge its referral to Dromaeosauridae. Our reanalysis of two distinct phylogenetic datasets focusing on basal paravian taxa supports the reinterpretation of *Balaur* as an avialan more crownward than *Archaeopteryx* but outside of Pygostylia, and as a flightless taxon within a paraphyletic assemblage of long-tailed birds. Our placement of *Balaur* within Avialae is not biased by character weighting. The placement among dromaeosaurids resulted in a suboptimal alternative that cannot be rejected based on the data to hand. Interpreted as a dromaeosaurid, *Balaur* has been assumed to be hypercarnivorous and predatory, exhibiting a peculiar morphology influenced by island endemism. However, a dromaeosaurid-like ecology is contradicted by several details of *Balaur*'s morphology, including the loss of a third functional manual digit, the non-ginglymoid distal end of metatarsal II, and a non-falciform ungual on the second pedal digit that lacks a prominent flexor tubercle. Conversely, an omnivorous ecology is better supported by *Balaur*'s morphology and is consistent with its phylogenetic placement within Avialae. Our reinterpretation of *Balaur* implies that a superficially dromaeosaurid-like taxon represents the enlarged, terrestrialised descendant of smaller and probably volant ancestors.

## INTRODUCTION

The theropod dinosaur *Balaur bondoc* from the Maastrichtian (latest Late Cretaceous) of Romania represents the most complete theropod dinosaur yet known from the Upper

Cretaceous of Europe (*Csiki et al., 2010*). The remarkably well-preserved holotype specimen of *B. bondoc*, EME (Transylvanian Museum Society, Dept. of Natural Sciences, Cluj-Napoca, Romania) PV.313, was collected from red overbank floodplain sediments of the Maastrichtian Sebeş Formation in 2009 and comprises an articulated partial postcranial skeleton of a single individual, including dorsal, sacral and caudal vertebrae as well as much of the pectoral and pelvic girdles and limbs (*Brusatte et al., 2013*). The first phylogenetic studies incorporating *Balaur* concluded that it represents an aberrant dromaeosaurid with velociraptorine affinities, endemic to the European palaeoislands of the Late Cretaceous (*Csiki et al., 2010*; *Turner, Makovicky & Norell, 2012*; *Brusatte et al., 2013*). The matrices utilised in these three studies have all been versions of the Theropod Working Group (TWiG) matrix, an incrementally and independently developed large-scale matrix focusing on the interrelationships of coelurosaurian taxa (e.g., *Norell, Clark & Makovicky, 2001*; *Makovicky, Apesteguía & Agnolín, 2005*; *Turner et al., 2007*; *Turner, Makovicky & Norell, 2012*; *Brusatte et al., 2014*). Comparison made between *Balaur* and other dromaeosaurids reveals the possession of a suite of autapomorphies not present in dromaeosaurids nor in most other non-avialan theropods, such as a fused carpometacarpus, loss of a functional third manual digit, proximal fusion of the tarsometatarsus, and a relatively enlarged first pedal digit (*Csiki et al., 2010*; *Brusatte et al., 2013*). Interpreted as a dromaeosaurid, *Balaur* is a strikingly odd and apparently avialan-like taxon. Recently, *Godefroit et al. (2013a)* included *Balaur* in a new phylogenetic analysis focusing on paravians and found it resolved as a basal avialan, more crownward than *Archaeopteryx*. A similar result was obtained independently by *Foth, Tischlinger & Rauhut (2014)* using a dataset expanded from that of *Turner, Makovicky & Norell (2012)*: *Foth, Tischlinger & Rauhut (2014)* recovered *Balaur* in a position relatively less crownward than in the tree obtained by *Godefroit et al. (2013a)*, but still crownward of *Archaeopteryx*. The present study focuses on resolving these conflicting interpretations regarding the affinities of *Balaur* following examination of the holotype material (performed by TB and DN). We also present a revised phylogenetic hypothesis based on a comparison of updated versions of previously published taxon-character matrices.

## MATERIALS AND METHODS

In order to test the competing dromaeosaurid and avialan hypotheses for the affinities of *Balaur*, we coded the holotype specimen into modified versions of two recently published theropod phylogenetic matrices: *Brusatte et al. (2014)* and *Lee et al. (2014)*. Both of these large-scale and independently coded matrices focused on the interrelationships of theropod dinosaurs and contain a broadly overlapping and comprehensive sampling of over 100 theropod taxa (152 and 120 taxa respectively), including many basal avialans. The two matrices differ from each other in the logical basis on character statement definitions (*Sereno, 2007*; *Brazeau, 2011*, see 'Discussion' below).

### *Brusatte et al. (2014)* data set

The dataset used by *Brusatte et al. (2014)* is an updated version of the dataset of *Turner, Makovicky & Norell (2012)*. We modified the *Brusatte et al. (2014)* matrix for this study

to include seven new characters and updated character states for three previously defined characters (see Supplemental Information). All character statements considered to be ordered by *Brusatte et al. (2014)* were set accordingly. The resulting data matrix (860 characters vs. 152 taxa) was then analysed using the Hennig Society version of TNT v1.1 (*Goloboff, Farris & Nixon, 2008*; see Supplemental Information for further details regarding modifications to the matrix and tree search strategy).

### *Lee et al. (2014)* data set

The dataset used by *Lee et al. (2014)* is an updated version of the dataset of *Godefroit et al. (2013a)*. Character statements of the 1,549 included characters and the source of scores for the included 120 fossil taxa are stored at the Dryad Digital Repository (*Cau et al., 2014*). In our study, this dataset has been expanded to including an additional taxonomic unit based on the extant avian *Meleagris* (ACUB 4817); accordingly, character statement 318 has been modified (see Supplemental Information). *Balaur* was re-scored based on our examination of the specimen and the incorporation of information from *Brusatte et al. (2013)*. *Lee et al. (2014)* applied Bayesian inference in their analysis of this dataset and integrating the morphological information with chronostratigrafic information. In the present study, the updated morphological data matrix (1,549 characters vs. 121 taxa) was analysed using parsimony as the tree search strategy in TNT (see Supplemental Information).

## Alternative placement test and implied weighting analyses

In our analyses of both datasets, we constrained the alternative deinonychosaurian and avialan positions for *Balaur*, measuring step changes between resultant topologies as a further indication of their relative support. Templeton's test (*Templeton, 1983*) was used to determine whether the step differences between the unforced and forced topologies were statistically significant. The backbone constraints used the following species: a crown avian (*Anas platyrhynchus* in the dataset of *Brusatte et al., 2014*, *Meleagris gallopavo* in the dataset of *Lee et al., 2014*), a dromaeosaurid (*Dromaeosaurus albertensis* in both datasets), and a troodontid (*Troodon formosus* in both datasets).

In order to test whether assumptions on downweighting of homoplasious characters influence the placement of *Balaur* among Paraves, both datasets were subjected to implied weighting analyses (IWAs, *Goloboff, 1993*; *Goloboff et al., 2008*; *Goloboff, Farris & Nixon, 2008*; see Supplemental Information).

## COMPARATIVE ANATOMY OF *BALAUR* AND OTHER MANIRAPTORAN THEROPODS

Compared to other theropods, *Balaur* displays a unique and unexpected combination of characters (*Brusatte et al., 2013*). The phylogenetic analyses of *Csiki et al. (2010)* and *Turner, Makovicky & Norell (2012)* resolved *Balaur* as a velociraptorine dromaeosaurid. Consequently, most of the unusual characters shared by *Balaur* with non-dromaeosaurid theropods were interpreted as autapomorphies, independently evolved along the lineage leading exclusively to *Balaur*. An alternative explanation is that these features may indicate a closer relationship between *Balaur* and another non-dromaeosaurid clade of

maniraptorans. Here, we list the most relevant characters that may support or challenge the alternative placements of *Balaur* within Maniraptora.

## Dorsal vertebrae with stalked parapophyses

The dorsal vertebrae of *Balaur* bear distinctly stalked parapophyses (*Brusatte et al., 2013*). Although this feature has been reported as a deinonychosaurian synapomorphy (*Turner, Makovicky & Norell, 2012*), stalked parapophyses are also present in alvarezsaurids and basal avialans (*Novas, 1997*; *Chiappe et al., 1999*; *Agnolín & Novas, 2013*).

## Sacrum including at least seven fused vertebrae

The presence of five fused sacral vertebrae is the plesiomorphic condition within coelurosaurs (e.g., *Brochu, 2003*). An independent increase in the number of fused sacral vertebrae is a widespread phenomenon within Maniraptoriformes. Six to seven sacral vertebrae are present in ornithomimids (*Osmólska, Roniewicz & Barsbold, 1972*), Late Cretaceous oviraptorosaurs (*Barsbold et al., 2000*), and some dromaeosaurids (*Norell & Makovicky, 1997*; *Turner, Makovicky & Norell, 2012*; S Brusatte, pers. comm., 2014). The synsacrum is composed of seven vertebrae in parvicursorine alvarezsauroids, whereas in basal taxa it includes only five vertebrae (*Choiniere et al., 2010*). *Archaeopteryx* and basal paravians retain five sacral vertebrae (*Hwang et al., 2002*; *Paul, 2002*; *Godefroit et al., 2013b*; *Godefroit et al., 2013a*), whereas a sacrum with at least seven vertebrae has been regarded as a synapomorphy of *Jixiangornis* and pygostylians (*Turner, Makovicky & Norell, 2012*). *Balaur* has at least seven sacral vertebrae: four fused and clearly discernible sacral vertebrae bearing sacral ribs are followed by three additional and co-ossified caudosacrals (*Brusatte et al., 2013*).

## Fused scapulocoracoid

In *Balaur*, the scapula and coracoid are co-ossified and the suture is obliterated on both sides (Fig. 1A; *Brusatte et al., 2013*). *Brusatte et al. (2013)* noted that a fused scapulocoracoid is present in some dromaeosaurids (e.g., *Adasaurus*, *Microraptor*, *Velociraptor*; see Fig. 1C) but not in others (e.g., *Achillobator*, *Buitreraptor*, *Deinonychus*, *Sinornithosaurus*, *Unenlagia*). *Turner, Makovicky & Norell (2012)* included fusion of the scapulocoracoid among the phylogenetically informative characters of their paravian phylogeny. Within non-avialan coelurosaurs, the presence of this character state has been reported within ornithomimosaurs, therizinosauroids, alvarezsauroids, tyrannosaurids and oviraptorosaurs (*Osmólska, Roniewicz & Barsbold, 1972*; *Perle, 1979*; *Perle et al., 1994*; *Brochu, 2003*; *Balanoff & Norell, 2012*), suggesting a high degree of homoplasy. Fusion of the scapulocoracoid is also present in basal avialans (e.g., Confuciusornithidae; *Chiappe et al., 1999*) and flightless avians (e.g., *Struthio*; ACUB 4820).

## Coracoid with prominent tuber placed on the anterolateral corner

The coracoid of *Balaur* bears a hypertrophied tubercle that forms the anterolateral corner of the bone and obscures the supracoracoid nerve foramen when the coracoid is observed in lateral view (Fig. 1A; *Brusatte et al., 2013*). Non-avialan theropods possess tubercles

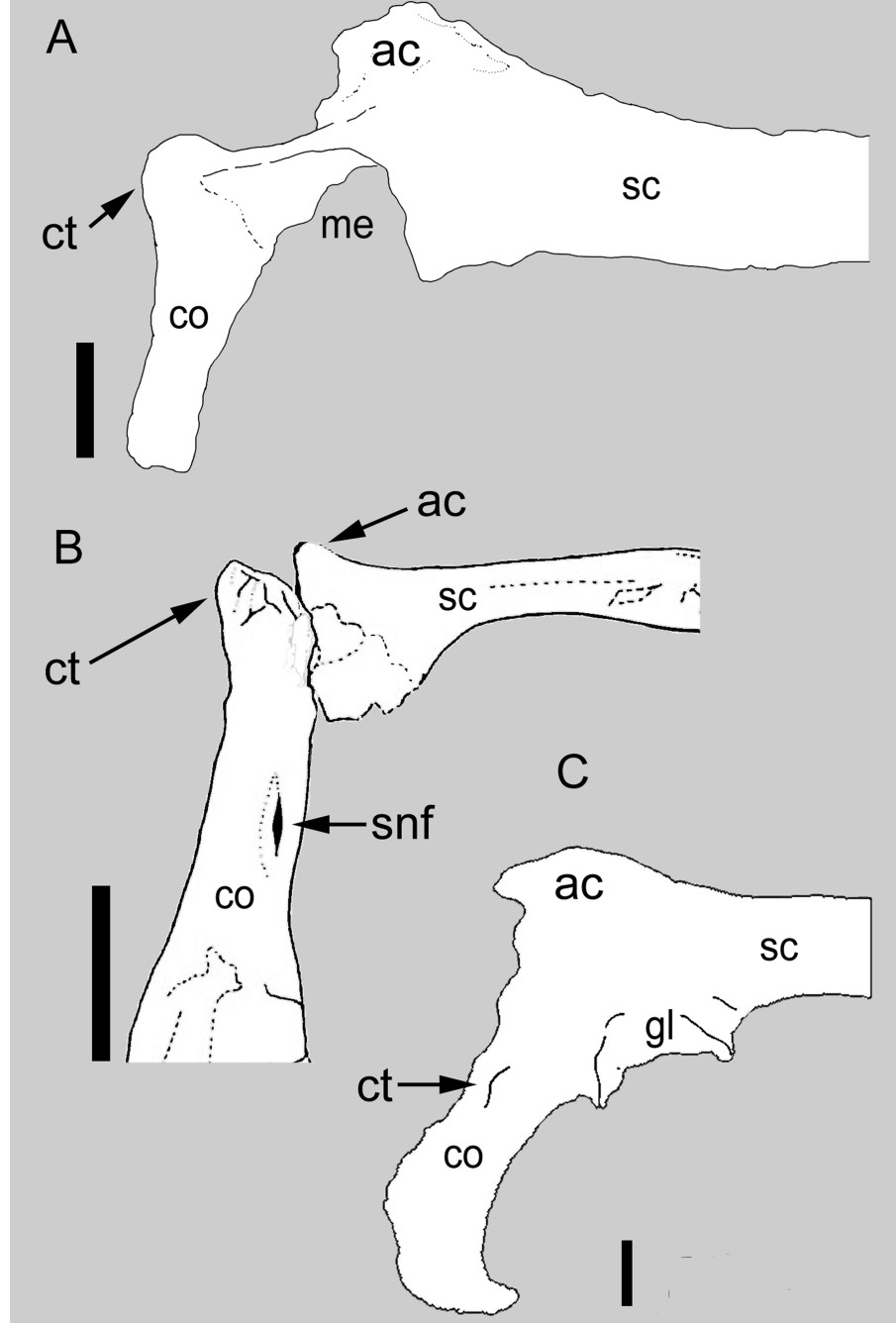

**Figure 1  Comparison between the scapulocoracoid of *Balaur* and other paravians.** Comparison of the scapulocoracoid of (A) *Balaur* (lateral view) to that of (B) the pygostylian *Enantiophoenix* (medial view); and (C) the dromaeosaurid *Velociraptor* (lateral view); (A) after *Csiki et al.*. (*2010*, Fig. 1); (B) modified after *Cau & Arduini* (*2008*, Fig. 2); (C) after *Norell & Makovicky* (*1999*, Fig. 4). All scapulocoracoids are drawn with the proximal half of the scapular blade oriented horizontally to show relative placement of coracoid tubercle. Scale bar: 10 mm (A); 5 mm (B); 10 mm (C). Abbreviations: ac, acromion; co, coracoid; ct, coracoid tubercle; gl, glenoid; me, missing element; sc, scapula; snf, supracoracoid nerve foramen.

that are relatively smaller and more lateroventrally directed (when the scapula is oriented horizontally) than that seen in avialan theropods (Fig. 1C; *Osmólska, Roniewicz & Barsbold, 1972*; *Ostrom, 1976*; this is the "*processus praeglenoidalis*" *sensu Elzanowski, Chiappe & Witmer, 2002*). Although the coracoid tubercle of *Balaur* may appear autapomorphic among non-avialan theropods (*Brusatte et al., 2013*), a prominent coracoid tubercle is also present in unenlagiines (*Buitreraptor*, see *Agnolín & Novas, 2013*), basal avialans (e.g., *Jeholornis, Jixiangornis*; *Turner, Makovicky & Norell, 2012*, Fig. 82), and forms the acrocoracoid of ornithothoracines (e.g., *Apsaravis, Enantiophoenix, Enantiornis*; *Clarke & Norell, 2002*; *Baier, Gatesy & Jenkins, 2007*; *Cau & Arduini, 2008*; *Walker & Dyke, 2009*; Fig. 1). A hypertrophied coracoid tubercle that obscures the supracoracoid nerve foramen in lateral view is also seen in *Sapeornis* (*Zhou & Zhang, 2003*; *Gao et al., 2012*).

## Humerus longer than half the combined length of tibiotarsus and tarsometatarsus

The ratio between the lengths of the humerus and femur is usually considered as a phylogenetically informative character in discussions on the evolution of coelurosaurian theropods (e.g., *Brusatte et al., 2014*, character 262), as that ratio is usually higher among avialans than it is in most non-avialan theropods. Since the femur of *Balaur* is unknown (*Brusatte et al., 2013*), we used the ratio between the length of the humerus and the sum of the lengths of the tibiotarsus and tarsometatarsus. The humerus of non-avialan theropods is consistently shorter than half the combined length of the tibiotarsus and tarsometatarsus (e.g., *Deinonychus, Gallimimus, Microraptor, Tyrannosaurus*; *Ostrom, 1969*; *Osmólska, Roniewicz & Barsbold, 1972*; *Hwang et al., 2002*; *Brochu, 2003*). In *Balaur*, the humerus is longer than half the combined length of the tibiotarsus and tarsometatarsus (55%) and approaches the condition seen in basal avialans (e.g., *Archaeopteryx*: 59%, *Confuciusornis*: 67%, *Jeholornis*: 77%; *Chiappe et al., 1999*; *Elzanowski, 2001*; *Zhou & Zhang, 2002*; see *Brusatte et al., 2013*, Table 2). The interpretation of this feature is problematic, since distal hindlimb elongation is not correlated to femur length among theropods (*Holtz Jr, 1995*); accordingly, we have avoided its inclusion among the new characters added to the phylogenetic analyses. We have retained the original humerus/femur ratio characters in both datasets and thus score *Balaur* as "unknown" for them. Therefore, the results of our analyses are not biased by the use of that character.

## Humeral condyles placed on the anterior surface of the distal end

The humerus of *Balaur* possesses condyles that are placed entirely on the anterior surface of the bone (*Brusatte et al., 2013*). As in *Balaur*, the complete anterior migration of the humeral condyles is present in therizinosauroids (e.g., *Zanno, 2010*), alvarezsauroids (*Novas, 1997*), basal pygostylians (e.g., *Confuciusornis, Limenavis, Enantiornis*; *Chiappe et al., 1999*; *Clarke & Chiappe, 2001*; *Walker & Dyke, 2009*), and extant birds (e.g., *Dromaius, Meleagris, Struthio*; ACUB 3131; 4817; 4820). All other known dromaeosaurids (e.g., *Deinonychus*; *Ostrom, 1969*), most non-avialan theropods (e.g., *Gallimimus, Allosaurus, Tyrannosaurus*; *Osmólska, Roniewicz & Barsbold, 1972*; *Madsen, 1976*; *Brochu, 2003*), and early avialans (e.g., *Archaeopteryx*; Berlin specimen) bear the condyles in a more

distal position, with a limited, if not absent, extent onto the anterior surface of the bone. In the analysis of *Turner, Makovicky & Norell (2012)*, *Balaur* was scored as retaining the primitive condition (*contra Brusatte et al., 2013*; *Brusatte et al., 2014*).

### Deep and elongate triangular brachial fossa on humerus

The humerus of *Balaur* bears a prominent triangular fossa on the anterior surface of the distal end of the humerus (*Brusatte et al., 2013*, Fig. 12). This fossa is bordered both laterally and medially by raised crests confluent with the epicondyles. The same configuration defines the brachial fossa present in pygostylians (e.g., *Confuciusornis*, *Limenavis*, *Apsaravis*; *Chiappe et al., 1999*; *Clarke & Chiappe, 2001*; *Clarke & Norell, 2002*). This fossa is also variably developed within dromaeosaurids (e.g., *Bambiraptor*; *Turner, Makovicky & Norell, 2012*; *Brusatte et al., 2013*).

### Ulna with brachial depression

The proximal third of *Balaur*'s ulna bears a shallow, elongate depression on the medial surface termed the "proximal fossa" (*Brusatte et al., 2013*, Fig. 14). This character is topographically equivalent to the brachial fossa present in pygostylians (*Baumel & Witmer, 1993*; *Clarke & Chiappe, 2001*; *Walker & Dyke, 2009*). The ulna of most non-avialan theropods lacks a brachial depression or possesses a poorly developed one (e.g., *Allosaurus*, *Tyrannosaurus*; *Madsen, 1976*; *Brochu, 2003*). However, the structure is well developed in some dromaeosaurids (e.g., *Bambiraptor*, *Buitreraptor*; *Burnham, 2004*; *Agnolín & Novas, 2011*; *Agnolín & Novas, 2013*).

### Distal carpals fused to proximal end of metacarpals

The manus of *Balaur* displays co-ossification of the distal carpals with the proximal ends of the metacarpals (Fig. 2A; *Brusatte et al., 2013*), unlike the dromaeosaurid condition in which no such fusion in present (Fig. 2D). The fusion between the distal carpals and the metacarpals is present in a few non-avialan theropod lineages (e.g., *Avimimus*, *Mononykus*; *Kurzanov, 1981*; *Perle et al., 1993*), and in pygostylians (e.g., *Confuciusornis*, *Xiangornis*; *Chiappe et al., 1999*; *Hu et al., 2012*). In particular, the pattern of proximal fusion among the carpometacarpal elements in *Balaur* is shared by most basal pygostylians (e.g., *Confuciusornis*, *Sinornis*, *Sapeornis*, *Pengornis*, *Enantiornis*, *Zhouornis*; *Chiappe et al., 1999*; *Sereno, Chenggang & Jianjun, 2002*; *Zhou & Zhang, 2003*; *Zhou, Clarke & Zhang, 2008*; *Walker & Dyke, 2009*; *Zhang et al., 2013*; see Figs. 2B–2C and Fig. S1). Most ornithurines and some enantiornithines display complete distal fusion between metacarpals II and III in addition to the aforementioned proximal fusion of the carpometacarpus as seen in *Balaur* (e.g., *Apsaravis*, *Teviornis*, *Xiangornis*; *Clarke & Norell, 2002*; *Kurochkin, Dyke & Karhu, 2002*; *Hu et al., 2012*).

### Semilunate carpal shifted laterally and first metacarpal sloped proximolaterally

In *Balaur*, the semilunate carpal overlaps the whole proximal ends of both metacarpals II and III (Fig. 2A and Fig. S1). Furthermore, the proximal end of the first metacarpal in

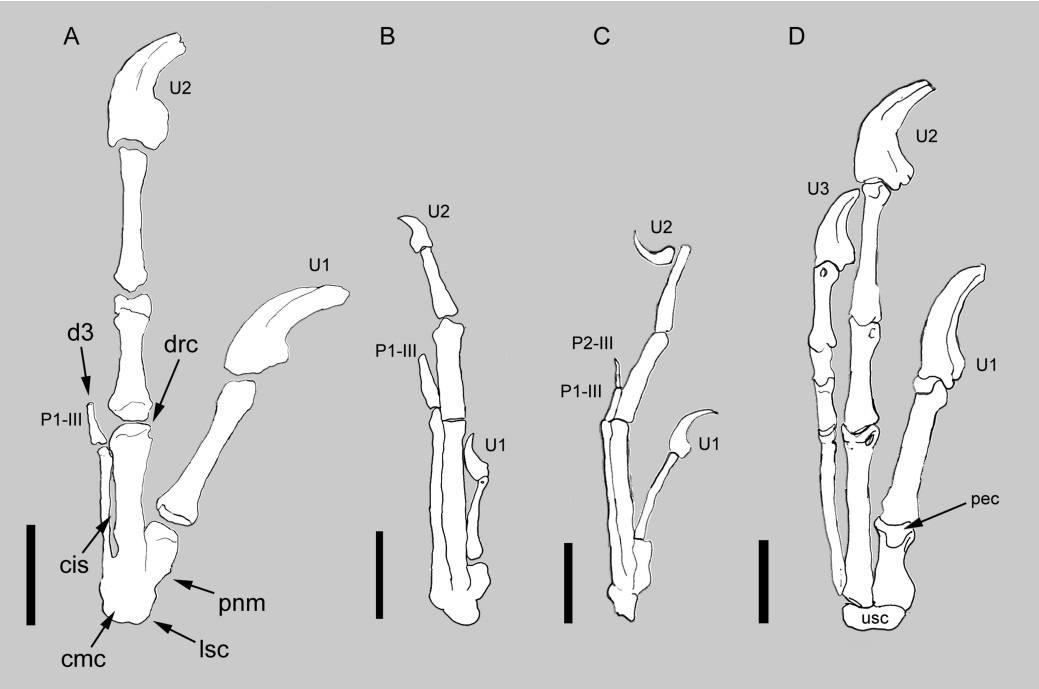

**Figure 2 Comparison between the manus of *Balaur* and other paravians.** Comparison of the manus of (A) *Balaur* to those of (B) the enantiornithine *Zhouornis*; (C) the pygostylian *Sapeornis*; and (D) the dromaeosaurid *Deinonychus*, showing bird-like features of *Balaur*. (A) after *Csiki et al.* (*2010*, Fig. 1, mirrored from original); (B) after *Zhang et al.* (*2013*, Fig. 7); (C) after *Zhou & Zhang* (*2003*, Fig. 7); (D) after *Wagner & Gauthier* (*1999*, Fig. 2). All drawn at the same metacarpal II length. Scale bar: 20 mm (A, D); 10 mm (B, C). Abbreviations: cis, closed intermetacarpal space; cmc, carpometacarpus; d3, reduced third digit; drc, distally restricted condyles; lsc, laterally shifted semilunate carpal; p1-III, first phalanx of manual digit 3; p2-III, second phalanx of manual digit 3; pec, proximally expanded extensor surface; pnm, proximally narrow metacarpal I; U, ungual; usc, unfused semilunate carpal.

*Balaur* is mediolaterally narrower than the distal end, producing a proximolaterally sloping medial margin of the metacarpus. In *Archaeopteryx* and most non-avialan maniraptorans, the proximal end of the first metacarpal is not constricted compared to the distal end, and the semilunate carpal overlaps most of metacarpal I; meanwhile, the overlap on metacarpal III is absent or limited to the medialmost margin of the bone (Fig. 2D; *Ostrom, 1976*, Fig. 10; *Xu, Han & Zhao, 2014*). Therefore, the position of the semilunate carpal of *Balaur* represents a lateral shift compared to the condition in other non-avialan maniraptorans, and recalls long-tailed and pygostylian birds where the semilunate carpal has a reduced or absent overlap on metacarpal I and extensively covers both metacarpals II and III (e.g., *Confuciusornis, Sinornis, Sapeornis, Enantiornis, Zhouornis*; *Chiappe et al., 1999*; *Sereno, Chenggang & Jianjun, 2002*; *Zhou & Zhang, 2003*; *Walker & Dyke, 2009*; *Zhang et al., 2013*; see also *Xu, Han & Zhao, 2014*; see Figs. 2B–2C). As in *Balaur*, pygostylian birds show a mediolateral constriction of the proximal end of the first metacarpal, and a medial margin ("anterior margin", using *Nomina Anatomica Avium* nomenclature, see *Harris, 2004*) that is variably sloped proximolaterally in extensor view.

## Condyles of metacarpals I–II restricted to the distal and ventral surfaces of the metacarpals

Metacarpals I and II of *Balaur* bear condyles that are restricted to the distal and ventral surfaces of the metacarpals, and are excluded from the extensor surfaces (*Brusatte et al., 2013*). The dromaeosaurid condition (e.g., *Deinonychus*, *Velociraptor*, *Graciliraptor*; *Ostrom, 1969*; *Norell & Makovicky, 1999*; *Xu & Wang, 2003*), in which the condyles are expanded along the extensor surface of the metacarpals, is present in most non-avialan theropods (e.g., *Acrocanthosaurus*, *Allosaurus*, *Australovenator*, *Berberosaurus*, *Dilophosaurus*, *Patagonykus*, *Rapator*; *Madsen, 1976*; *Welles, 1984*; *Novas, 1997*; *Senter & Robins, 2005*; *Allain et al., 2007*; *White et al., 2013*). The condition present in the metacarpals of *Balaur* is also present in pygostylians (e.g., *Teviornis*, *Sinornis*, *Enantiornis*; *Kurochkin, Dyke & Karhu, 2002*; *Sereno, Chenggang & Jianjun, 2002*; *Walker & Dyke, 2009*) and extant avians (e.g., *Dromaius*; *Meleagris*, *Struthio*; ACUB 3131; 4817; 4820). Furthermore, the ventral surface of the metacarpals of *Balaur* are excavated by a wide flexor sulcus but lack distinct flexor pits at the distal end, similar to the condition present in avialans (e.g., *Teviornis*; *Kurochkin, Dyke & Karhu, 2002*) but differing from that of dromaeosaurids and most non-avialan theropods that do bear a distinct flexor pit (e.g., *Allosaurus*, *Acrocanthosaurus*, *Mahakala*, *Velociraptor*; *Madsen, 1976*; *Senter & Robins, 2005*; *Turner, Pol & Norell, 2011*).

## Metacarpal II with an intermetacarpal ridge running along the dorsolateral edge of the bone and closed intermetacarpal space between metacarpals II and III

*Balaur* possesses a distinct web of bone that extends along the dorsolateral edge of metacarpal II and contacts metacarpal III distally, and a distally closed intermetacarpal space between metacarpals II and III (*Brusatte et al., 2013*). Within basal avialans, the extent of the contact between metacarpals II and III displays some variation, ranging from the close contact of a straight metacarpal III to metacarpal II with no intermetacarpal space (e.g., *Sapeornis*; *Zhou & Zhang, 2003*; *Gao et al., 2012*; see Fig. 2 and Fig. S1), an appressed distal contact but no fusion of metacarpal III to metacarpal II (the condition as seen in *Balaur* and many basal avialans, including *Jeholornis*, *Enantiornis*, *Confuciusornis*, *Zhouornis*, and *Piscivoravis*; *Zhou & Zhang, 2002*; *Walker & Dyke, 2009*; *Zhang et al., 2009*; *Zhang et al., 2013*; *Zhou, Zhou & O'Connor, 2014*), to distal obliteration of the contact between metacarpals II and III due to complete fusion between the bones (e.g., *Teviornis*, *Xiangornis*, *Meleagris*; *Kurochkin, Dyke & Karhu, 2002*; *Hu et al., 2012*; ACUB 4817). A closed intermetacarpal space is present in *Confuciusornis* (*Chiappe et al., 1999*; *Zhang et al., 2009*), some long-tailed birds (e.g., *Jeholornis*, *Jixiangornis*; *Zhou & Zhang, 2002*), and ornithothoracines (e.g., *Enantiornis*, *Xiangornis*, *Zhouornis*; *Walker & Dyke, 2009*; *Hu et al., 2012*; *Zhang et al., 2013*; see Fig. 2B). Most euornithines differ from *Balaur* and most other avialans in having a more distally placed intermetacarpal space relative to a more shortened metacarpal I (e.g., *Teviornis*; *Kurochkin, Dyke & Karhu, 2002*).

## Distal end of metacarpal III unexpanded and not divided into separated condyles

The third metacarpal of *Balaur* bears a simple distal end that lacks distinct condyles. Dromaeosaurids share with most non-avialan theropods the presence of well-defined metacarpal condyles separated by an intercondylar sulcus (e.g., *Allosaurus, Bambiraptor, Deinocheirus, Deinonychus, Dilophosaurus, Gallimimus*; *Ostrom, 1969*; *Osmólska & Roniewicz, 1970*; *Osmólska, Roniewicz & Barsbold, 1972*; *Madsen, 1976*; *Welles, 1984*; *Burnham, 2004*). The condition present in the third metacarpal of *Balaur* is shared by tyrannosaurids (e.g., *Tyrannosaurus*; *Lipkin & Carpenter, 2008*, Fig. 10.10), basal pygostylians (e.g., *Confuciusornis, Enantiornis, Sinornis, Teviornis, Xiangornis, Zhouornis*; *Chiappe et al., 1999*; *Kurochkin, Dyke & Karhu, 2002*; *Sereno, Chenggang & Jianjun, 2002*; *Walker & Dyke, 2009*; *Hu et al., 2012*; *Zhang et al., 2013*) and crown avians (e.g., *Meleagris, Struthio*; ACUB 4817; 4820). This character is not obviously linked with the reduction in the number of phalanges in digit III (see below), since *Confuciusornis* shows the derived metacarpal condition (i.e., simple distal end of metacarpal III) yet retains a full set of four functional phalanges in digit III.

## Third manual digit bearing less than three phalanges

The third manual digit of *Balaur* is extremely reduced and lacks the distal phalanges, including the ungual (Fig. 2A; *Brusatte et al., 2013*). The only known phalanx in the third manual digit of *Balaur* has a tapering distal end with a small distal articular surface, suggesting the presence of a possible additional phalanx of very small size. Such a reduction is unknown in dromaeosaurids, which have three non-ungual phalanges on manual digit III and a fully functional ungual (Fig. 2D), but it is commonly found in non-confuciusornithid pygostylians, where the third manual digit is usually reduced to two or fewer phalanges, the most distal of which has a tapering distal end and poorly defined articular surfaces (e.g., *Sinornis, Sapeornis, Zhouornis, Piscivoravis*; *Sereno, Chenggang & Jianjun, 2002*; *Gao et al., 2012*; *Zhang et al., 2013*; *Zhou, Zhou & O'Connor, 2014*; see Figs. 2B, 2C and Fig. S1).

## Dorsal margin of manual unguals does not arch dorsally above level of articular facet and flexor tubercles not expanded ventrally

*Senter (2007a)* argued that in dromaeosaurid manual unguals, the dorsal margins arch higher than the articular facets when the latter is held vertically, and that this feature differentiates dromaeosaurid manual unguals from those of other theropods. The derived condition is present in microraptorines and eudromaeosaurs but is absent in unenlagiines (*Senter, 2007a*; *Senter, 2007b*; *Currie & Paulina Carabajal, 2012*; Figs. S1A and S1B). Furthermore, the manual unguals in both dromaeosaurids and troodontids bear prominent and dorsoventrally expanded flexor tubercles. Following the method described by *Senter (2007a)*, we note that the dorsal margins of *Balaur*'s manual unguals do not arch higher than the articular facet, and that the flexor tubercles are relatively low and more elongate proximodistally than they are deep dorsoventrally (*Brusatte et al., 2013*, Figs. 21 and 22; Fig. S1C). Similar absence of a markedly convex dorsal margin of the manual

ungual and relatively moderate development of the flexor tubercles is widespread among the manual unguals of basal avialans (e.g., *Sinornis*, *Sapeornis*, *Zhouornis*, *Piscivoravis*; *Sereno, Chenggang & Jianjun, 2002*; *Gao et al., 2012*; *Zhang et al., 2013*; *Zhou, Zhou & O'Connor, 2014*; see Figs. 2B and 2C).

## Complete coossification of pelvic bones

*Balaur* displays coossification of the pelvic bones such that both the iliopubic and ilioischial sutures are obliterated (*Brusatte et al., 2013*, Fig. S2A). In most tetanuran theropods, including basalmost avialans, the pelvic elements do not completely coossify (e.g., *Allosaurus*, *Jeholornis*, *Patagonykus*, *Sapeornis*, *Tyrannosaurus*; *Madsen, 1976*; *Novas, 1997*; *Zhou & Zhang, 2002*; *Brochu, 2003*; *Zhou & Zhang, 2003*). This contrasts with ceratosaurian-grade theropods (*Tykoski & Rowe, 2004*), some non-avialan coelurosaurs (e.g., *Avimimus*; *Kurzanov, 1981*) and ornithothoracines (e.g., *Apsaravis*, cf. *Enantiornis*, *Patagopteryx*, *Qiliania*, *Sinornis*; *Chiappe, 2002*; *Chiappe & Walker, 2002*; *Clarke & Norell, 2002*; *Sereno, Chenggang & Jianjun, 2002*; *Ji et al., 2011*, Fig. S2D) in which the pelvic bones fuse completely. Although coossification of the ilium to the pubis is present in the only known specimen of the microraptorine dromaeosaurid *Hesperonychus*, the pelvic coossification differs from *Balaur* and avialans as the ilioischial articulation remains unfused (*Longrich & Currie, 2009*).

## Ridge bounding the cuppedicus fossa confluent with the acetabular rim

In the ilium of *Balaur*, the ridge that dorsally bounds the cuppedicus fossa is extended posteriorly on the lateral surface of the pubic peduncle and is confluent with the acetabular rim (*Brusatte et al., 2013*; Fig. S2A). This feature is a compound character formed by the presence of a ridge bounding the cuppedicus fossa, which is a neotetanuran synapomorphy (*Hutchinson, 2001*; *Novas, 2004*), and the posterior extension of the cuppedicus fossa on the lateral surface of the pubic peduncle, which is a derived feature of paravians (*Hutchinson, 2001*, Figs. 4–6). The combination of features present in *Balaur* is shared by *Anchiornis* and *Xiaotingia* (*Turner, Makovicky & Norell, 2012*), *Unenlagia* and *Rahonavis* (*Novas, 2004*), *Velociraptor* (*Norell & Makovicky, 1999*) and enantiornithines (e.g., *Sereno, Chenggang & Jianjun, 2002*, Fig. 8.4; *Walker & Dyke, 2009*, Fig. S2D). The presence and extent of the cuppedicus fossa is difficult to determine in most Mesozoic avialans because of the two-dimensional preservation of most specimens (*Novas, 2004*). Furthermore, the character statements relative to the ridge bounding the cuppedicus fossa in phylogenetic analyses are marked as 'inapplicable' in those taxa lacking a distinct cuppedicus fossa (*Hutchinson, 2001*; e.g., *Mahakala*, *Patagopteryx*, Ornithurae; *Turner, Pol & Norell, 2011*), a scoring strategy followed by both *Turner, Makovicky & Norell (2012)* and *Godefroit et al. (2013a)*.

## Pubis and ischium projected strongly posteroventrally and subparallel

*Balaur* has a posteroventrally directed pubis, subparallel to the ischium (*Csiki et al., 2010*; Fig. S2A). Although *Brusatte et al. (2013)* acknowledged that the extreme posterior

inclination of the pubis may partially be the result of taphonomic distortion, they confirmed the genuine posteroventral orientation of this bone. Within Theropoda, retroversion of the pubis (opisthopuby) is known in therizinosauroids, parvicursorine alvarezsaurids, dromaeosaurids and pygostylians. Most therizinosauroids (but not *Falcarius*) show a posteroventrally directed pubis that articulates with the obturator process of the ischium (*Zanno, 2010*). Opisthopuby is present in many parvicursorines (e.g., *Mononykus*; *Perle et al., 1994*), but absent in more basal alvarezsauroids (e.g., *Haplocheirus*, *Patagonykus*; *Novas, 1997*; *Choiniere et al., 2010*). A retroverted pubis is absent in basal paravians—they instead display a vertically oriented ('mesopubic') pubis—and is present in some dromaeosaurids (e.g., *Adasaurus* and *Velociraptor*; *Norell & Makovicky, 1999*; *Xu et al., 2010*; *Turner, Makovicky & Norell, 2012*) but absent in others (e.g., *Achillobator*, *Utahraptor*; *Perle, Norell & Clark, 1999*; *Senter et al., 2012*). It is also a common feature in pygostylian birds (e.g., *Confuciusornis*, *Patagopteryx*, *Sapeornis*; *Chiappe et al., 1999*; *Hutchinson, 2001*; *Chiappe, 2002*; *Chiappe & Walker, 2002*; *Zhou & Zhang, 2003*).

## Broad pelvic canal with laterally convex pubes and abrupt distal narrowing of interpubic distance

*Brusatte et al. (2013)* noted as an autapomorphy of *Balaur* an interpubic distance that is proportionally greater than that present in other dromaeosaurids (e.g., *Velociraptor*; *Norell & Makovicky, 1997*; *Norell & Makovicky, 1999*). The gap between the laterally bowed pubes of *Balaur* only begins to narrow abruptly in the distalmost third of the bone (Fig. 3B and Fig. S2B; *Brusatte et al., 2013*, Fig. 56). This condition differs from that seen in most theropods (e.g., *Avimimus*, *Sinraptor*, *Tyrannosaurus*; *Currie & Zhao, 1993*; *Vickers-Rich, Chiappe & Kurzanov, 2002*; *Brochu, 2003*), including *Velociraptor* (Fig. 3D and Fig. S2C; *Norell & Makovicky, 1999*; *Brusatte et al., 2013*), *Bambiraptor* (*Burnham, 2004*) and *Archaeopteryx* (*Norell & Makovicky, 1999*, Fig. 25), where the narrowing is more gradual over the length of the pubes and the pubis is not bowed laterally in anteroposterior view. *Brusatte et al. (2013)* noted that the condition in *Balaur* is somewhat similar to the condition in therizinosaurids (*Zanno, 2010*). The combination of a relatively broad pelvic canal, bounded by laterally convex pubes and with an abrupt distal narrowing of the interpubic distance, is also seen in pygostylian birds (e.g., *Concornis*, *Dapingfangornis*, *Piscivoravis*, *Sapeornis*, *Yanornis*; *Sanz, Chiappe & Buscalioni, 1995*; *Zhou & Zhang, 2003*; *Li et al., 2006*; *Zhou, Zhou & O'Connor, 2014*; *Zheng et al., 2014*; see Fig. 3C and Fig. S2E).

## Ischial tuberosity

The ischium of *Balaur* bears a well-developed obturator tuberosity (*ischial tuberosity* of *Hutchinson, 2001*) on the dorsal end of the part of its anterior margin that contacts or nearly contacts the pubis ventrally (*Brusatte et al., 2013*). This feature was determined to be a synapomorphy of the velociraptorine subclade (including *Balaur*) by *Turner, Makovicky & Norell (2012)*. However, almost all non-velociraptorine taxa were scored by them as either unknown for or lacking an ischial tuberosity (char. 176 in *Turner, Makovicky & Norell, 2012*), with only *Adasaurus*, *Anchiornis*, *Deinonychus* and *Velociraptor* scored as bearing that feature. Nevertheless, a prominent ischial tuberosity is also present in avialans,

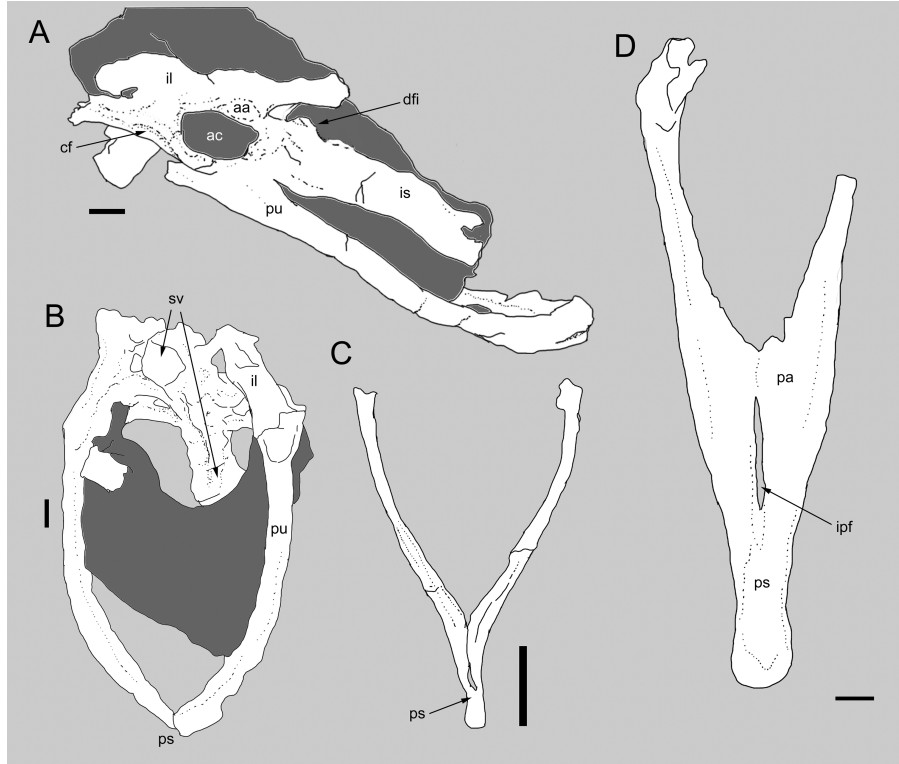

**Figure 3 Comparison between the pelvis of *Balaur* and other paravians.** Pelvis of *Balaur* in lateral view (A). Comparison of the pubes of *Balaur* in anteroventral view (B) to those of the pygostylian *Sapeornis* in anterior view (C), and the dromaeosaurid *Velociraptor* in posterior view (D). (C) after *Zhou & Zhang* (*2003*, Fig. 8); (D) after *Norell & Makovicky* (*1999*, Fig. 19). Scale bar: 10 mm (A, B, D), 2 mm (C). Abbreviations: aa, antitrochanter; ac, acetabulum; cf, cuppedicus fossa; dfi, dorsal flange of ischium; ipf, interpubic fenestra; is, ischium; pa, pubic apron; ps, pubic symphysis; pu, pubis, sv, sacral vertebrae.

in particular in large-bodied flightless taxa (e.g., *Patagopteryx*; *Hutchinson, 2001*). The ischial tuberosity of some avialans approaches and contacts the pubis (e.g., *Dromaius*; ACUB 3131), and is the case in *Balaur*.

## Ischium with proximodorsal flange

The ischium of *Balaur* bears a process along the dorsal half of its dorsal surface (*Brusatte et al., 2013*, Fig. 27A: "dorsal flange of proximal ischium"). This process is topographically equivalent to the proximal dorsal ischial tuberosity of other reptiles (*Hutchinson, 2001*). This structure is variably developed on the ischia of many paravians (e.g., *Novas & Puerta, 1997*; *Forster, 1998*; *Xu, Wang & Wu, 1999*; *Agnolín & Novas, 2013*). In unenlagiines and microraptorines, the ischium bears a tuber-like proximodorsal process (*Novas & Puerta, 1997*; *Agnolín & Novas, 2013*, Fig. 3.5 C–E) which is absent in known velociraptorines (*Norell & Makovicky, 1999*; *Agnolín & Novas, 2013*; *Brusatte et al., 2013*) except for a *Velociraptor*-like taxon from Mongolia (*Norell & Makovicky, 1999*, Fig. 24). In basal avialans, the ischial tuberosity is developed as a prominent trapezoidal flange which is more proximodistally expanded than it is in other paravians and which resembles the condition present in *Balaur* (e.g., *Confuciusornis*, cf. *Enantiornis*, *Jeholornis*, *Patagopteryx*,

*Sapeornis, Sinornis*; *Chiappe et al., 1999*; *Hutchinson, 2001*; *Sereno, Chenggang & Jianjun, 2002*; *Zhou & Zhang, 2002*; *Zhou & Zhang, 2003*; *Walker & Dyke, 2009*; see *Agnolín & Novas, 2013*; Figs. S2D and S2F).

### Fibula fused to tibia proximally

In *Balaur*, the tibia and the fibula are fused proximally (*Brusatte et al., 2013*), a condition not seen in dromaeosaurids or most non-avialan theropods. Among coelurosaurs, a more extensive proximal fusion between tibia and fibula is present in pygostylian birds (e.g., *Qiliania*; *Ji et al., 2011*).

### Tuber and ridge along lateral surface of the distal end of the tibiotarsus

The distal end of the tibiotarsus of *Balaur* bears a pronounced anteroposteriorly oriented lateral ridge. The ridge is most pronounced anteriorly, where it terminates at a discrete rounded tubercle located at the point where the lateral condyle and shaft merge. The ridge is kinked at its midpoint where it forms a second, ventrally directed tubercle positioned laterodistally relative to the first tubercle (*Brusatte et al., 2013*, Fig. 35). *Brusatte et al. (2013)* suggested that the first tubercle may represent the distal end of the fibula, fused to the tibiotarsus, whereas no interpretation of the second tubercle was provided. A raised ridge along the anterolateral margin of the distal end of the tibiotarsus at the point of fusion between the tibia and the proximal tarsals is also present in *Qiliania* (*Ji et al., 2011*) and in the enigmatic Haţeg taxon *Bradycneme* (*Harrison & Walker, 1975*). Based on comparison with birds, we interpret the second tubercle and the corresponding kinked ridge as the fibular facet of the calcaneum. According to our interpretation, the other (more proximally placed) tubercle is topographically equivalent to the *tuberculum retinaculi M. fibularis* of birds (*Baumel & Witmer, 1993*).

### Complete distal co-ossification of the tibiotarsus

The distal end of the tibia and the proximal tarsals of *Balaur* are coossified, forming a tibiotarsus where the sutures are obliterated (*Brusatte et al., 2013*). *Turner, Makovicky & Norell (2012)* considered the fusion between the calcaneum and astragalus, but not the tibia and tarsals, to be a synapomorphy of Paraves. Fusion involving the proximal tarsals and the distal end of the tibia is a condition seen in some basal neotheropods (*Tykoski & Rowe, 2004*). Within non-avialan coelurosaurs, coossification of the proximal tarsals and the distal end of the tibia is observed in alvarezsaurids (e.g., *Albinykus, Mononykus*; *Perle et al., 1994*; *Nesbitt et al., 2011*) and some oviraptorosaurs (e.g., *Avimimus, Elmisaurus*; *Osmólska, 1981*; *Vickers-Rich, Chiappe & Kurzanov, 2002*. Within Avialae, the presence of a fully coossified tibiotarsus is present in taxa more crownward than *Archaeopteryx* (e.g., *Apsaravis, Confuciusornis, Hollanda*; *Chiappe et al., 1999*; *Clarke & Norell, 2002*; *Bell et al., 2010*).

### Deep extensor groove on distal tibiotarsus

*Balaur* bears a deep and prominent extensor groove on the distal end of the tibiotarsus (*Brusatte et al., 2013*). Within dromaeosaurids, this feature has otherwise been

reported only in *Buitreraptor* and is homoplastically present in other maniraptoran lineages (e.g., *Apsaravis, Hollanda, Mononykus*; *Perle et al., 1994*; *Clarke & Norell, 2002*; *Bell et al., 2010*).

## Tibiotarsus with intercondylar sulcus extended along the posterior surface

The distal end of *Balaur*'s tibiotarsus is saddle-shaped due to the presence of a large and distinct intercondylar sulcus (*Brusatte et al., 2013*). The latter feature is restricted not only to the anterodistal end of the bone but also extends along the distal end of the posterior surface as a flexor sulcus. This feature is also present in basal avialans known from three-dimensionally preserved specimens (e.g., *Apsaravis, Hollanda*; *Clarke & Norell, 2002*; *Bell et al., 2010*).

## Deep circular pit on medial surface of distal tibiotarsus

The medial surface of the distal end of *Balaur*'s tibiotarsus is excavated by a deep subcircular pit which was described as being deeper than are the homologous depressions variably present in the astragali of some dromaeosaurids (*Brusatte et al., 2013*). A pit comparable in depth to that present in *Balaur* is also present in avialans more crownward than *Archaeopteryx* (*depressio epicondylaris medialis*, *Baumel & Witmer, 1993*) and has been considered a phylogenetically informative feature (see *O'Connor, Chiappe & Bell, 2011*).

## Extensive coossification of tarsometatarsus

The tarsometatarsal elements of *Balaur* display extensive coossification (Fig. 4A, Figs. S3 and S4; *Brusatte et al., 2013*), in contrast to most non-avian theropods in which no such fusion is present (e.g., *Velociraptor*; see Fig. 4B and Fig. S4A). Many maniraptoran lineages display coossification of the distal tarsals to the proximal ends of the metatarsals (e.g., *Avimimus, Adasaurus, Albinykus, Elmisaurus*; *Kurzanov, 1981*; *Osmólska, 1981*; *Nesbitt et al., 2011*; *Turner, Makovicky & Norell, 2012*). However, the extensive coossification of the metatarsal shafts is a character present only in *Balaur* and pygostylians (e.g., *Bauxitornis, Confuciusornis, Evgenavis, Hollanda, Patagopteryx, Vorona, Yungavolucris*; *Chiappe, 1993*; *Chiappe et al., 1999*; *Chiappe, 2002*; *Forster et al., 2002*; *Bell et al., 2010*; *Dyke & Ösi, 2010*; *O'Connor, Averianov & Zelenkov, 2014*; see Fig. 4C, Figs. S3, S4C and S4D).

## Metatarsals with one or more longitudinal eminences on the dorsal surface of the shafts

The shafts of *Balaur*'s second to fourth metatarsals are dorsoventrally deep in cross-section, being strongly convex along the extensor surfaces except for the area of contact between metatarsals II and III. Here, the lateral edge of metatarsal II and the medial edge of metatarsal III form dorsoventrally shallow, longitudinally arranged flanges that, together, form a depressed region between the remainder of the metatarsal shafts. This unusual character combination, which is not observed in non-avialan theropods, was considered to be an autapomorphy of *Balaur* by *Brusatte et al. (2013)*. However, comparable features are present in several Mesozoic avialans. *Vorona* possesses two distinct ridges that extend along the distal halves of the extensor surfaces of both metatarsals III and IV, delimiting a

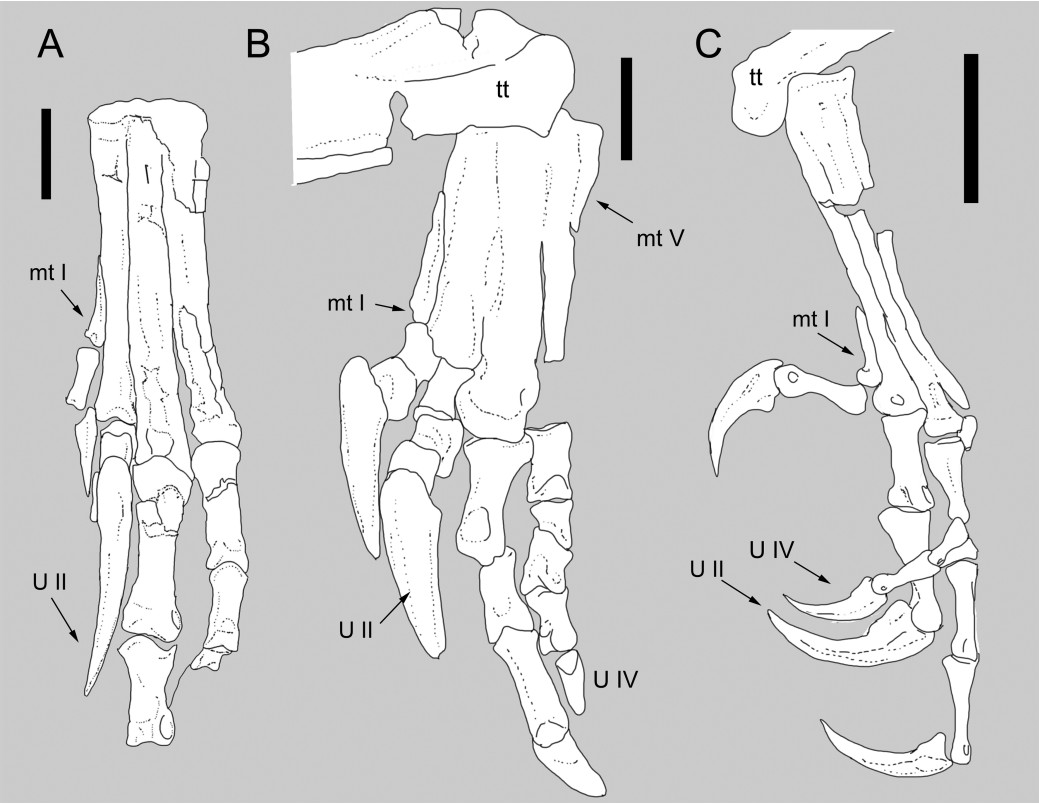

**Figure 4 Comparison between the metatarsus of *Balaur* and other paravians.** Comparison of the metatarsus and pes of (B) *Balaur* to that of (A) the dromaeosaurid *Velociraptor*; and (C) the pygostylian *Zhouornis*. (A) after *Norell & Makovicky* (*1997*, Figs. 6); (C) after *Zhang et al.*. (*2013*, Fig. 8, mirrored from original). Scale bar: 20 mm (A, B); 10 mm (C). Abbreviations: mt I, metatarsal I; mt V, metatarsal V; tt, tibiotarsus; U II: pedal ungual II; U IV, pedal ungual IV.

depressed intermetatarsal space (*Forster et al., 2002*). A depressed area between metatarsals II and III is also present in *Patagopteryx* (*Chiappe, 2002*). The extensor surfaces of metatarsals II and III are markedly convex transversely in many avisaurids with depressed areas present between the metatarsal shafts (e.g., *Avisaurus*, *Bauxitornis*; *Chiappe, 1993*; *Dyke & Ösi, 2010*; Fig. S3H). *Yungavolucris* is reported to lack a dorsally convex third metatarsal; however, the shaft's extensor surface at the proximal end of metatarsal III bears a centrally positioned, longitudinally oriented eminence comparable to the condition in *Balaur* (*Chiappe, 1993*). Finally, the enigmatic avialan *Mystiornis* also bears distinct longitudinal ridges along the extensor surfaces of metatarsals II–IV (*Kurochkin et al., 2010*).

### Enlarged extensor fossa on distal end of metatarsal II

In most theropods, the distal end of metatarsal II bears an extensor fossa proximal to the articular end. This fossa usually appears as a pit delimited by distinct margins and does not extend mediolaterally across the entire extensor surface (e.g., *Allosaurus*, *Deinonychus*, *Tyrannosaurus*; *Ostrom, 1969*; *Madsen, 1976*; *Brochu, 2003*). In *Balaur*, the extensor fossa of metatarsal II is enlarged and extends across the whole distal surface, bounded

laterally by a raised ridge converging with the trochlea (*Brusatte et al., 2013*; Fig. S3B). A large, proximodistally enlarged extensor fossa is present on the second metatarsal of *Evgenavis* (*O'Connor, Averianov & Zelenkov, 2014*; Fig. S3F). An enlarged extensor fossa on metatarsal II, lacking distinct margins and bounded laterally by a raised margin, is also present in *Parabohaiornis* (*Wang, O'Connor & Zhou, 2014*) and *Yungavolucris* (*Chiappe, 1993*; Fig. S3E).

## Metatarsal II with plantarly projected medial condyle

*Balaur* bears a plantarly projected medial condyle on the distal end of metatarsal II, visible in medial view as a distinct ventral projection of the distal end (*Brusatte et al., 2013*; Fig. S3A). In most theropods, including dromaeosaurids, the medial condyle of metatarsal II does not project plantarly more than the lateral condyle (e.g., *Deinonychus, Eustreptospondylus, Falcarius, Garudimimus, Sinraptor, Talos, Tyrannosaurus, Zuolong*; *Ostrom, 1969*; *Currie & Zhao, 1993*; *Brochu, 2003*; *Kirkland et al., 2004*; *Kobayashi & Barsbold, 2005*; *Sadleir, Barrett & Powell, 2008*; *Choiniere et al., 2010*; *Zanno et al., 2011*). Many avialans bear a plantarly unexpanded medial condyle on metatarsal II and hence resemble other theropods (e.g., *Avisaurus, Mystiornis, Yungavolucris*; *Chiappe, 1993*; *Kurochkin et al., 2010*). However, a plantarly projected medial condyle like that present in *Balaur* is present in the basal pygostylians *Confuciusornis* and *Evgenavis* (*O'Connor, Averianov & Zelenkov, 2014*; Figs. S3G and S3I) and in the ornithuromorph *Apsaravis* (*Clarke & Norell, 2002*).

## Metatarsal II lacks prominent ginglymoid distal end

The presence of a prominent extensor sulcus on the second metatarsal is regarded as a synapomorphy of Dromaeosauridae (*Turner, Makovicky & Norell, 2012*). *Balaur* possesses a broadly convex distal end of metatarsal II that lacks a ginglymoid distal articulation with a well-developed extensor sulcus (Fig. 4A; see *Norell & Makovicky, 1997*; *Brusatte et al., 2013*; Fig. S3B). Some avialan taxa also bear a distinct extensor sulcus on metatarsal II like that present in dromaeosaurids (e.g., *Avisaurus, Yungavolucris*; *Chiappe, 1993*; Figs. S3C and SE) whereas others bear a broadly convex articular facet and hence resemble *Balaur* (e.g., *Bauxitornis, Evgenavis*; *Dyke & Ösi, 2010*; *O'Connor, Averianov & Zelenkov, 2014*; Figs. S3D and S3F).

## Distal articular surface of metatarsal II narrower than maximum width of its distal end

The width of the distal articular surface of metatarsal II in *Balaur* is less than the width of the entire distal end of the metatarsal (*Brusatte et al., 2013*; Fig. S3B). In extensor view, a large non-articular region is present both lateral and medial to the articular surface. The metatarsals of most therizinosauroids show a similar condition (e.g., *Segnosaurus*; *Perle, 1979*). The same feature also occurs in the second metatarsal of some avisaurid avialans, where distinct non-articular mediolateral expansions are present proximal to the distal articular surface (*Avisaurus archibaldi, A. gloriae*; *Chiappe, 1993*; *Varricchio & Chiappe, 1995*; Fig. S3C).

### Shaft of metatarsal IV anteroposteriorly compressed and mediolaterally widened

In most theropods, the mid-length cross section of metatarsal IV is subcircular, or anteroposteriorly thicker than wide. In *Balaur*, the mid-length cross section of metatarsal IV is anteroposteriorly compressed and mediolaterally expanded (*Brusatte et al., 2013*), a characteristic that is also seen in both velociraptorine (e.g., *Deinonychus*, *Velociraptor* and *Adasaurus*) and dromaeosaurine dromaeosaurids (e.g., *Utahraptor*), as well as basal troodontids (*Turner, Makovicky & Norell, 2012*). However, an anteroposteriorly compressed metatarsal IV with a flat cross section is also present in basal avialans (e.g., *Avisaurus*, *Mystiornis*, *Evgenavis*, *Yungavolucris*; *Brett-Surman & Paul, 1985*; *Chiappe, 1993*; *Kurochkin et al., 2010*; *O'Connor, Averianov & Zelenkov, 2014*; Figs. S3E and S3H).

### Short and robust metatarsal V

Dromaeosaurids bear a slender and elongate metatarsal V that is at least 40% of metatarsal III's length (Fig. 5C; *Norell & Makovicky, 1999*; *Hwang et al., 2002*; *Brusatte et al., 2013*). *Balaur* possesses a shorter and stouter metatarsal V that is less than 30% of metatarsal III's length (Fig. 5A, Figs. S3A and S4B, ; *Brusatte et al., 2013*): it is thus more similar to the condition present in basal avialans (e.g., *Evgenavis*, *Sapeornis*, *Vorona*; *Forster et al., 2002*; *Zhou & Zhang, 2003*; *O'Connor, Averianov & Zelenkov, 2014*) and most non-avialan coelurosaurs (e.g., *Khaan*, *Segnosaurus*, *Tyrannosaurus*; *Perle, 1979*; *Brochu, 2003*; *Balanoff & Norell, 2012*).

### Hallux unreduced compared to other toes and functional

*Balaur* possesses a hallux that cannot be considered reduced in size compared to the other pedal digits (*Brusatte et al., 2013*, Fig. S4B). Most non-avialan theropods, including dromaeosaurids, possess a relatively small first pedal ungual (e.g., *Allosaurus*, *Microraptor*, *Velociraptor*; *Madsen, 1976*; *Norell & Makovicky, 1997*; *Hwang et al., 2002*; Fig. S4A). However, a large and falciform first pedal ungual that is not reduced compared to the other pedal unguals, as seen in *Balaur*, is also present in many basal avialans (e.g., *Confuciusornis*, *Jixiangornis*, *Patagopteryx*, *Sapeornis*, *Zhouornis*; *Chiappe et al., 1999*; *Chiappe, 2002*; *Ji et al., 2002*; *Zhou & Zhang, 2003*; *Zhang et al., 2013*; Fig. S4D). Furthermore, the first phalanx in *Balaur*'s hallux is subequal in length compared to the proximal phalanges of pedal digits II–IV, a condition present in basal avialans (e.g., *Jixiangornis*, *Sapeornis*, *Zhouornis*; *Ji et al., 2002*; *Zhou & Zhang, 2003*; *Zhang et al., 2013*; Fig. S4) but not in non-avialan theropods. The distal placement of the articular end of metatarsal I in *Balaur* relative to the trochlea of metatarsal II is more similar to that of basal avialans (e.g., *Confuciusornis*, *Patagopteryx*; *Chiappe et al., 1999*; *Chiappe, 2002*) than the more proximally placed trochlea of metatarsal I in dromaeosaurids (e.g., *Microraptor*, *Deinonychus*, *Velociraptor*; *Norell & Makovicky, 1997*; *Hwang et al., 2002*; *Fowler et al., 2011*) and other non-avialan theropods (e.g., *Khaan*, *Balanoff & Norell, 2012*). In addition, the well-developed articular surfaces indicate that the hallux of *Balaur* was dextrous, mobile and fully functional (*Brusatte et al., 2013*). This is also the condition present in birds but contrasts with that of most non-avialan theropods, including dromaeosaurids (*Norell & Makovicky, 1997*).

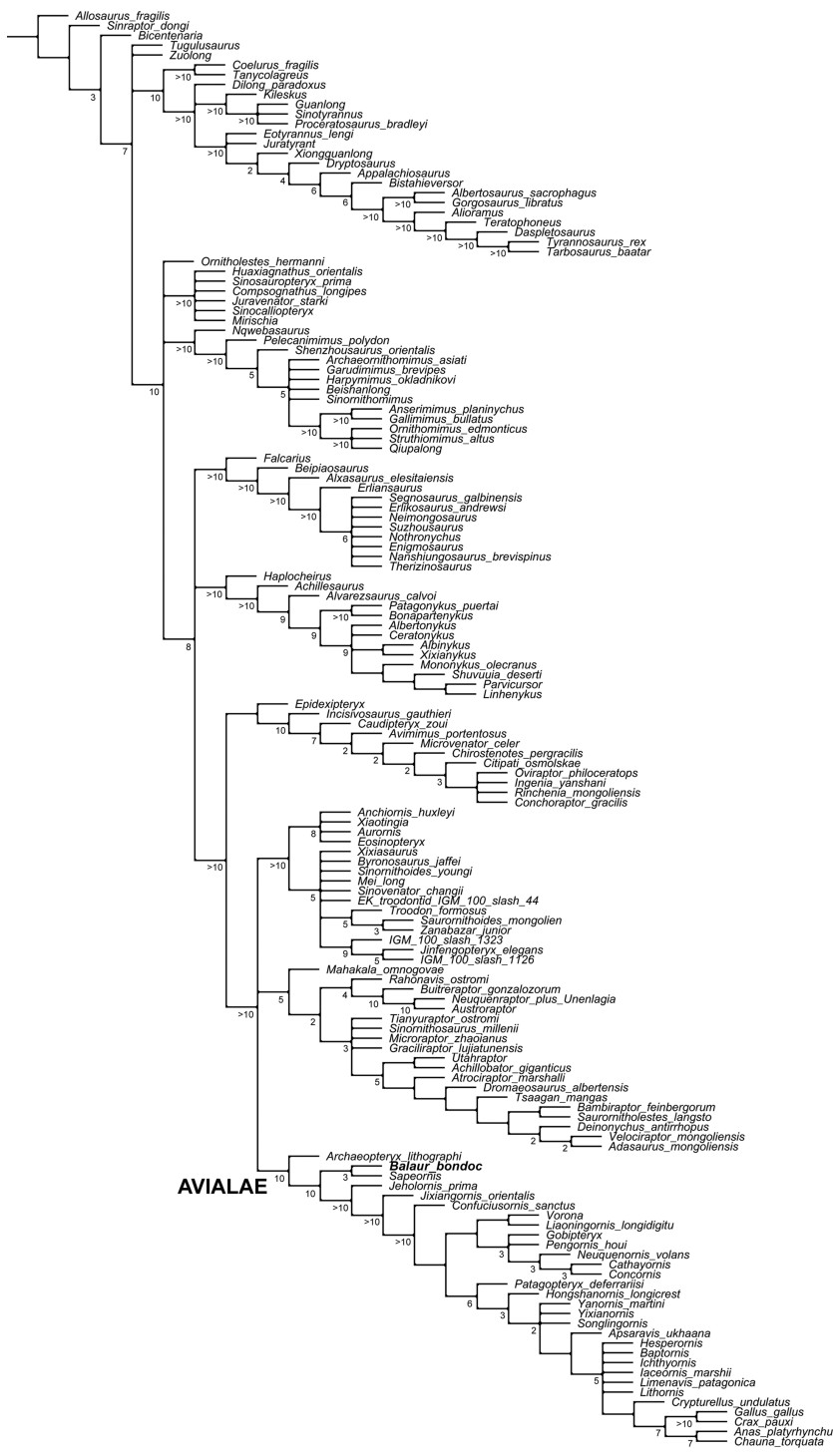

**Figure 5 Updated dataset of *Brusatte et al. (2014)*.** Reduced strict consensus of the shortest trees from the analysis of the modified *Brusatte et al. (2014)* matrix after pruning the 'wildcard' taxa *Epidendrosaurus*, *Hesperonychus*, *Kinnareemimus*, *Pedopenna*, *Pyroraptor*, and *Shanag*. Numbers adjacent to nodes indicate Decay Index values >1.

### Enlarged pedal ungual II lacking both marked falciform shape and prominent flexor tubercle

*Balaur* bears a hypertrophied second pedal ungual that is larger than the third and fourth pedal unguals, similar to that seen in most deinonychosaurs (*Turner, Makovicky & Norell, 2012*; *Brusatte et al., 2013*). However, *Brusatte et al. (2013)* noted that the second pedal ungual of *Balaur* does not show the marked falciform shape and prominent flexor tubercle seen in most dromaeosaurids (e.g., *Ostrom, 1969*; *Turner, Makovicky & Norell, 2012*). A robust second pedal digit with an enlarged and moderately recurved ungual, comparable to the condition in *Balaur*, is also present among several avialans (e.g., *Bohaiornis*, *Fortunguavis*, *Jixiangornis*, *Parabohaiornis*, *Patagopteryx*, *Qiliania*, *Sulcavis*, *Zhouornis*; *Chiappe, 2002*; *Ji et al., 2002*; *Hu et al., 2011*; *Ji et al., 2011*; *O'Connor et al., 2013*; *Zhang et al., 2013*; *Wang et al., 2014*; *Wang, O'Connor & Zhou, 2014*; Fig. S4D).

### Penultimate phalanges of pedal digit III more than 1.2 times longer than preceding phalanx

In most theropods, including dromaeosaurids, the penultimate phalanx of the third pedal digit is subequal to or shorter than the length of the preceding phalanges (e.g., *Gallimimus*, *Khaan*, *Tyrannosaurus*, *Velociraptor*; *Osmólska, Roniewicz & Barsbold, 1972*; *Norell & Makovicky, 1997*; *Brochu, 2003*; *Balanoff & Norell, 2012*; *Brusatte et al., 2013*, Table 7). However, *Balaur* bears a relatively elongate penultimate phalanx on pedal digit III that is 1.2 times longer than the preceding phalanx (*Brusatte et al., 2013*; Fig. S4B). This condition is similar to that present in many avialans (e.g., *Concornis*, *Sapeornis*, *Zhouornis*; *Sanz, Chiappe & Buscalioni, 1995*; *Zhou & Zhang, 2003*; *Zhang et al., 2013*; Fig. S4D) and unlike that of dromaeosaurids and most non-avialan theropods.

### Pedal ungual IV reduced in size

*Balaur*'s fourth pedal ungual, although distally incomplete in the holotype specimen, is the smallest of the pedal unguals (about 60% the size of pedal ungual III, see *Brusatte et al., 2013*; Fig. S4B). This condition differs from that seen in dromaeosaurids, *Sapeornis* and some troodontids in which the fourth pedal unguals are more than 85% the length of the third pedal ungual (e.g., *Borogovia* 140%, *Sapeornis* 100%; *Osmólska, 1987*; *Brusatte et al., 2013*, Table 7; *Pu et al., 2013*). It does, however, resemble the condition in ornithothoracine birds (e.g., *Bohaiornis* 59%, *Parabohaiornis* 60%, *Qiliania* 76%, *Zhouornis* 66%; *Hu et al., 2011*; *Ji et al., 2011*; *Zhang et al., 2013*; Figs. S4C and S4D). The relative length of the fourth pedal ungual in most maniraptorans is intermediate between *Balaur* and dromaeosaurids, being 70 and 85% the length of pedal ungual III (e.g., *Archaeopteryx* 77–78%, *Khaan*, *Jixiangornis*, *Sinornithoides* and *Zhongjianornis* 80%; *Elzanowski, 2001*; *Currie & Zhiming, 2001*; *Ji et al., 2002*; *Zhou & Li, 2010*; *Balanoff & Norell, 2012*, Fig. 33).

## RESULTS

### Modified *Brusatte et al. (2014)* analysis

The modified *Brusatte et al. (2014)* analysis produced >999,999 shortest cladograms of 3,397 steps each (CI = 0.3206, RI = 0.7771). In all the shortest trees found, *Balaur*

was recovered as an avialan, as the sister-taxon of *Sapeornis*, and not as a member of Dromaeosauridae. The '*Balaur* + *Sapeornis*' clade resolved as the sister-taxon of a clade including Pygostylia, *Jixiangornis* and *Jeholornis*. Exploration of the alternative topologies that we recovered resulted in *Epidendrosaurus*, *Hesperonychus*, *Kinnareemimus*, *Pedopenna*, *Pyroraptor*, and *Shanag* acting as 'wildcard' taxa among Maniraptora, and these taxa were pruned *a posteriori* from the results of the analyses to improve resolution within paravian taxa. After pruning the 'wildcard' taxa from the strict consensus topology (Fig. 5), *Archaeopteryx* was resolved as the sister-group to the rest of Avialae. Unambiguous synapomorphies for the sister taxon relationships between *Balaur* and *Sapeornis* are: anterior surface of deltopectoral crest with distinct muscle scar near lateral edge along distal end of crest for insertion of biceps muscle (139.1, homoplastic among maniraptorans); third manual digit with two or less phalanges (147.2, convergently developed among ornithothoracines); humeral condyles placed on anterior surface (366.1, convergently developed among therizinosauroids, alvarezsauroids and most avialans); anteroposterior diameter of metacarpal III less than 50% of the same diameter in metacarpal II (386.1); and length of first phalanx of pedal digit I greater than 66% of first pedal phalanx of pedal digit III (859.1, convergently developed among more crownward avialans).

Furthermore, all three versions of the dataset that used implied weighting recovered *Balaur* as an avialan more crownward than *Archaeopteryx* (see Fig. S5).

### Modified *Lee et al. (2014)* analysis

The modified *Lee et al. (2014)* analysis recovered 1,152 shortest trees of 6,350 steps each (CI = 0.2672, RI = 0.5993). The strict consensus of the shortest trees found is in general agreement with the Maximum Clade Credibility Tree recovered by *Lee et al. (2014)*, the most relevant difference being the unresolved polytomy among *Aurornis*, *Jinfengopteryx*, Dromaeosauridae, Troodontidae and Avialae (Fig. 6). The *a posteriori* pruning of the above mentioned genera does not resolve the polytomy among the three suprageneric clades. It is noteworthy that an unresolved polytomy among the main paravian lineages was also obtained by *Brusatte et al. (2014)*, and by our updated version of the latter dataset. In all trees found, *Balaur* was resolved as a basal avialan and as the sister taxon of Pygostylia (the '*Zhongjianornis* + (*Sapeornis* + more crownward avialans)' clade), in agreement with the results of previous versions of this matrix (i.e., *Godefroit et al., 2013b*). The character states unambiguously supporting this placement for *Balaur* are: (1) presence of fusion between metacarpal II and the distal carpals (char. 311.1); presence of a mediolaterally slender third metacarpal (char. 322.1); absence of the mediodorsal process on the ischium (char. 423.0); presence of an elongate first phalanx of pedal digit I (char. 499.0); and presence of a completely fused tibiotarsus (char. 580.1). Nodal support for this placement was low (Decay Index = 1). However, higher nodal support values for nodes along the less crownward part of Avialae support the placement of *Balaur* in this clade. This interpretation is further supported by the implied weighting analyses of the data set: these analyses consistently recovered *Balaur* as a non-pygostylian member of Avialae, located less

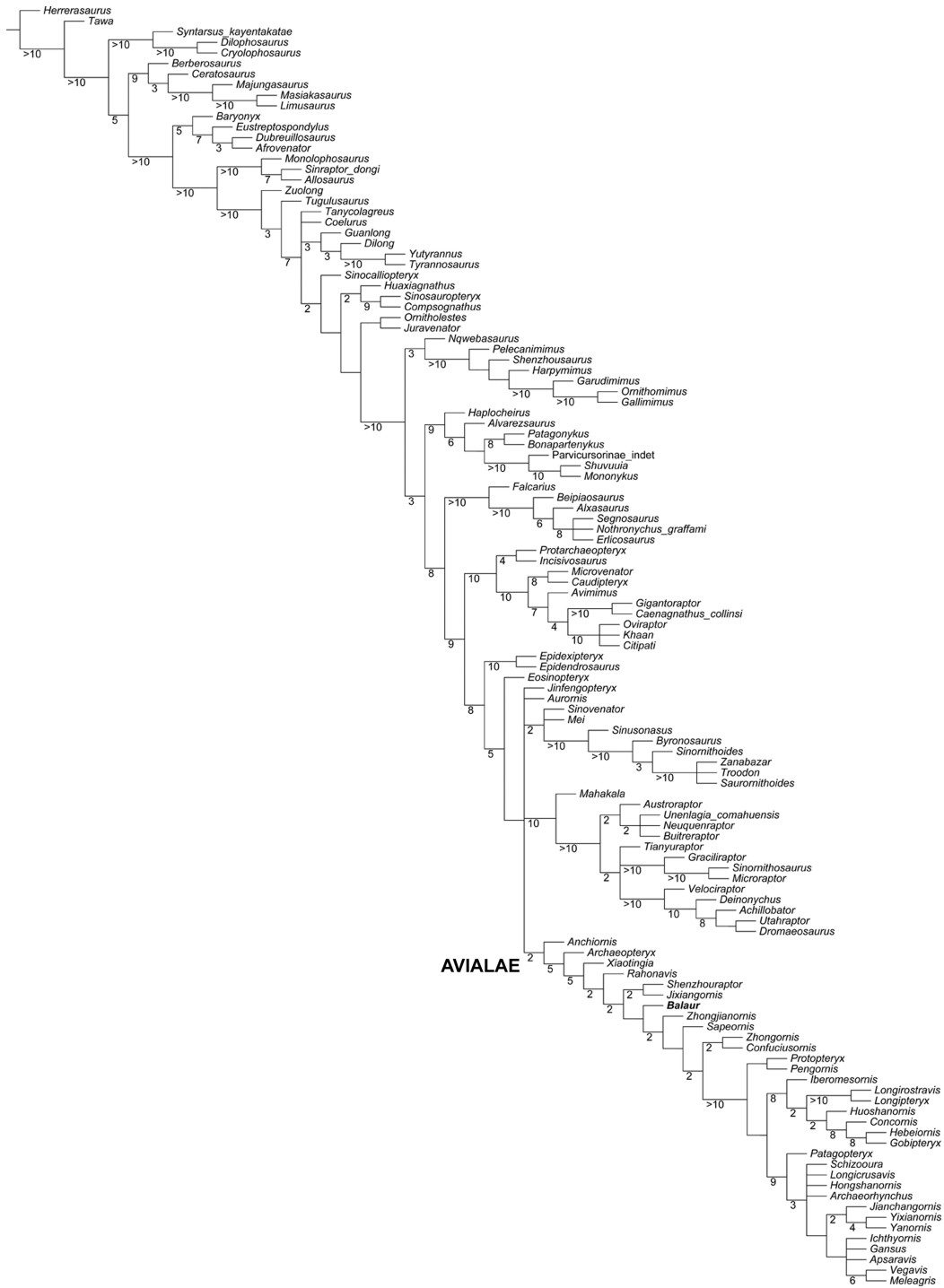

**Figure 6 Updated dataset of *Lee et al. (2014)*.** Strict consensus of the shortest trees from the analysis of the modified *Lee et al. (2014)* matrix. Numbers adjacent to nodes indicate Decay Index values > 1.

crownward within Avialae than was the case in the unweighted analysis, and bracketed by *Archaeopteryx* and all other avialans (see Fig. S6 ).

## Templeton tests

We re-analysed the original dataset of *Brusatte et al.*'s (*2014*) enforcing the following backbone constraint: ((*Balaur*, *Anas*), *Troodon*, *Dromaeosaurus*) (i.e., enforcing the analysis to retain only those topologies where *Balaur* is closer to modern birds than to either troodontids and dromaeosaurids, thus by definition forcing it to be a member of Avialae; see Supplemental Information). The shortest enforced topologies that resulted were 3,368 steps long, eight steps less parsimonious than the shortest unconstrained topologies that recovered *Balaur* among Dromaeosauridae. This difference was not statistically significant based on the Templeton test ($p > 0.2059, N > 37$).

We also re-analysed the modified dataset of *Brusatte et al. (2014)*, this time enforcing *Balaur* to be a dromaeosaurid using the following backbone constraint: ((*Balaur*, *Dromaeosaurus*), *Troodon*, *Anas*). The shortest enforced topologies that resulted were 3,400 steps long, three steps less parsimonious than the shortest unconstrained topologies where *Balaur* was recovered within Avialae. This difference was not statistically significant based on the Templeton test ($p > 0.7098, N > 61$).

Using the dataset modified from *Lee et al. (2014)*, we enforced a dromaeosaurid placement for *Balaur*, using the following backbone constraint: ((*Balaur*, *Dromaeosaurus*), *Troodon*, *Meleagris*). The shortest trees found using that constraint are nine steps longer than the shortest unforced topologies, and placed *Balaur* as the basalmost dromaeosaurid, excluded from the ((Eudromaeosauria + Microraptoria) + Unenlagiinae) clade. This difference was not statistically significant based on the Templeton test ($p > 0.4400, N > 125$).

Finally, we also tested a velociraptorine placement for *Balaur*, using the following backbone constraint: ((*Balaur*, *Velociraptor*), *Dromaeosaurus*, *Troodon*). The shortest trees found using that constraint are 14 steps longer than the shortest unforced topologies, and placed *Balaur* as the basalmost velociraptorine. This difference was not statistically significant based on the Templeton test ($p > 0.1580, N > 89$).

## DISCUSSION

*Balaur* possesses a unique and bizarre mix of characters, many of which were previously considered exclusive to Deinonychosauria or Avialae, and which may challenge its placement in either of the aforementioned clades. *Godefroit et al. (2013b)* tested alternative placements of *Balaur* among Paraves, and recovered the dromaeosaurid placement for that taxon as a suboptimal solution. Here, we have shown that an avialan placement for *Balaur* using the original dataset of *Brusatte et al. (2014)* is a suboptimal solution that cannot be rejected using that dataset. Although the most parsimonious results of the two updated phylogenetic analyses presented here concur in resolving *Balaur* within Avialae, the deinonychosaurian placement for this taxon discussed by *Brusatte et al. (2013)* can be only tentatively rejected based on current information. The most parsimonious placement was recovered under both equally weighted and implied weighting analyses, suggesting that the avialan placement of *Balaur* was not biased by *a priori* assumptions on homoplasious

character downweighting. Nevertheless, whatever the placement for *Balaur*, a significant amount of homoplasy, due to both convergences and reversals, is required to explain its unique morphology.

The sister taxon relationship recovered between *Balaur* and the short-tailed *Sapeornis*—resulting from analysis of the dataset modified from *Brusatte et al. (2014)*—is quite unexpected, and may be partially biased by the placement of the long-tailed *Jeholornis* and *Jixiangornis* as closer to other short-tailed birds than *Sapeornis* (a relationship also recovered by the original dataset, *Brusatte et al., 2014*). According to that topology, the short pygostyle-bearing tail of *Sapeornis* evolved independently of the same condition in more crownward birds. The topology that results from our use of the dataset modified from *Lee et al. (2014)* agrees with most analyses of avialan relationships (e.g., *Cau & Arduini, 2008*; *O'Connor, Chiappe & Bell, 2011*; *O'Connor et al., 2013*; *Wang et al., 2014*) in depicting a single origin of the pygostylian tail among birds.

Here we should note that topological discrepancies and alternative placements of problematic taxa may be influenced by artefacts in coding practice, or by the logical basis of character statement definition followed by different authors (*Brazeau, 2011*). The datasets of *Brusatte et al. (2014)* and *Lee et al. (2014)* differ from each other in the logical basis of their respective character statements and definitions. The definitions of many characters used in the analysis of *Brusatte et al. (2014)* impose congruence by linking more than one variable character to a particular state (see *Brazeau, 2011* and references therein), or by mixing together neomorphic and transformational characters as alternative states of the same character statements (see *Sereno, 2007*). For example, the ordered character 178 of *Brusatte et al. (2014)* includes three transformational states describing alternative extensions of the pubic apron along the pubis, together with a fourth state describing a distinct phenomenon, the absence of the pubic apron (a neomorphic character). Set that way, the absence of the pubic apron is *a priori* forced as a highly derived (and much weighted) terminal state of a character describing a feature (the proximodistal extent of the apron) that cannot be determined in those taxa lacking the apron. Character statements and definitions in the analysis of *Lee et al. (2014)* followed the recommendations outlined by *Sereno (2007)* and *Brazeau (2011)*; consequently, each character statement describes a single variable character, and neomorphic and transformational characters were included as separate character statements. To avoid the creation of spurious transformational optimizations under some topologies, the characters in the analysis of *Lee et al. (2014)* were therefore atomized in such a way as to capture both the presence or absence of the feature in addition to the states of the feature (*Brazeau, 2011*). Taxa scored as lacking a particular neomorphic character were scored as 'unknown' for the transformational characters describing different conditions of the same neomorphic feature.

We therefore consider it likely that some discrepancies between the updated analyses of *Brusatte et al. (2014)* and *Lee et al. (2014)*—including the alternative placements of *Balaur* and *Sapeornis* among basal avialans—reflect artefacts of coding rather than actual conflict in the data. Nevertheless, it is noteworthy that even using distinct datasets, alternative character weighting hypotheses and different logical bases for character definitions, *Balaur*

was consistently recovered as a basal avialan. Furthermore, the phylogenetic analysis of *Foth, Tischlinger & Rauhut (2014)*, which used the dataset of *Turner, Makovicky & Norell (2012)* as their basis and which included an expanded set of characters, independently found *Balaur* to be a basal avialan more crownward than *Archaeopteryx*, but in a less crownward position than that presented here. In conclusion, we consider the consensus among the results of these alternative tests (i.e., *Balaur* as a non-pygostylian basal avialan) as the phylogenetic framework for the discussion on its evolution and palaeoecology.

### Implications for the palaeoecology of *Balaur*

In the absence of both extrinsic data on diet and craniodental remains there is no direct evidence pertaining to the ecology and trophic adaptations of *Balaur*. Although not explicitly stated, *Brusatte et al.*'s (*2013*) inferences regarding the ecology and diet of *Balaur* rest entirely on their favoured phylogenetic placement of the taxon within the predatory deinonychosaurian clade Velociraptorinae (see *Carpenter, 1998*).

However, some aspects of *Balaur*'s morphology do not support the hypothesis that its ecomorphology was similar to that of dromaeosaurids. While there exists evidence that dromaeosaurids employed both their hands and feet in predation (see *Carpenter, 1998*), the reduction in length and functionality of the third manual digit and the poor development or absence of the pedal characters linked with predatory behaviour in deinonychosaurs (i.e., ginglymoid distal end of metatarsal II allowing extensive hyperextension, falciform second ungual with prominent flexor tubercle; *Ostrom, 1969*; *Fowler et al., 2011*), challenge the notion of a specialised, dromaeosaurid-like predatory ecology for *Balaur*. *Brusatte et al. (2013)* interpreted these unusual traits of *Balaur* as the result of insularism, although they acknowledged that comparable morphological changes in insular taxa have so far not been reported in predatory species. We are not aware of the reduction or loss of predatory adaptations in any insular predatory taxon, and therefore consider it unlikely that the unique morphology of *Balaur*, in particular the appendicular characters considered to be predatory adaptations among dromaeosaurids, could be sufficiently accounted for by the 'island effect'.

Most of the features considered to be autapomorphies of *Balaur* by *Csiki et al. (2010)* and *Brusatte et al. (2013)* are reinterpreted here as avialan synapomorphies. Consequently, these traits were inherited by *Balaur* from its bird-like ancestors before its lineage was isolated in the Hațeg environment. Since our analyses place *Balaur* among a grade of non-predatory avialans including herbivorous and/or omnivorous species (*Zhou & Zhang, 2002*; *Dalsätt et al., 2006*; *Zanno & Makovicky, 2011*), our preferred scenario does not necessitate a hypothesis of a carnivorous ecology for this taxon and is thus more consistent with the absence of the aforementioned predatory adaptations. Furthermore, in assuming a herbivorous or omnivorous ecology for *Balaur*, the amount of morphological changes, particularly in limb shapes and proportions, is comparable to that reported in several insular herbivorous and omnivorous taxa, including both mammals (*Sondaar, 1977*; *Caloi & Palombo, 1994*; *Van der Geer et al., 2011*) and dinosaurs (e.g., *Dalla Vecchia, 2009*). In particular, the presence in *Balaur* of a relatively broad pelvic canal, the short and broad

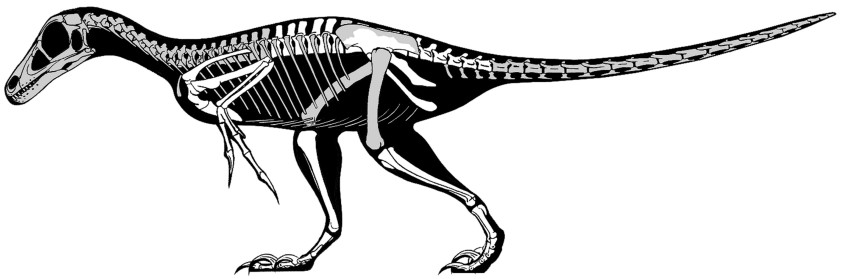

**Figure 7 Skeletal reconstruction of *Balaur*.** Speculative skeletal reconstruction for *Balaur bondoc*, showing known elements in white and unknown elements in grey. Note that the integument would presumably have substantially altered the outline of the animal in life. Produced by Jaime Headden, used with permission.

metatarsus with mediolaterally expanded distal ends relative to the articular surfaces, and the presence of an enlarged first pedal digit is a combination of features convergently acquired only by the non-predatory clade Therizinosauridae among Mesozoic theropods (*Zanno, 2010*; *Zanno & Makovicky, 2011*).

However, we agree with previous authors that, regardless of its position within Paraves, the morphology of *Balaur* includes a unique and unexpected combination of features, otherwise seen in distinct maniraptoran lineages. Interestingly, *Balaur* independently evolved a series of features previously reported in more crownward bird lineages, such as a deep *depressio epicondylaris medialis* in the tibiotarsus, a hypertrophied extensor fossa in the second metatarsal, and dorsally convex metatarsals with expanded distal ends (characters elsewhere seen in some ornithothoracines). A possible role of insularism in the origin of some of these traits is acknowledged even in our preferred phylogenetic scenario. In particular, the results of our analyses indicate that *Balaur* is phylogenetically bracketed by taxa showing relatively more elongate forelimbs (humeral lengths usually more than 60% of the tibiotarsus + tarsometatarsus length) and more robust forearms (ulna as thick as or thicker than the tibiotarsus). Accordingly, we interpret the forelimb of *Balaur* as secondarily reduced. Flightlessness has also been inferred in the ornithurine *Gargantuavis* from the Campanian-Maastrichtian of southern France (*Buffetaut & Loeuff, 1998*), indicating that distinct avialan lineages endemic to Late Cretaceous Europe reduced or lost their flight adaptations. Several bird clades independently evolved flightlessness during the Cenozoic as a result of their exploitation of insular environments and the taxa concerned typically displayed apomorphic reduction of the forelimbs compared to those of their closest relatives (*Paul, 2002*; *Naish, 2012*). Therefore, the reduced forelimb of *Balaur* may be interpreted as the result of insularism.

Finally, existing skeletal and life reconstructions of *Balaur* have interpreted it as a velociraptorine-like dromaeosaurid (*Csiki et al., 2010*; *Brusatte et al., 2013*). Does our re-interpretation of this taxon as a member of Avialae require that previous hypotheses about its appearance should be modified? By combining the known elements of *Balaur* with those of other paravians, a new skeletal reconstruction has been produced (Fig. 7). As our knowledge of Mesozoic paravian diversity has improved, it has become ever clearer

that early members of the deinonychosaurian and avialan lineages were highly similar in proportions, detailed anatomy and life appearance: consequently, an 'avialan interpretation' of *Balaur* does not result in an animal obviously different from a 'dromaeosaurid interpretation'. This conclusion has been supported by recent quantitative analyses that demonstrate a significant degree of shared morphospace between basal avialan taxa and their closest paravian relatives (e.g., *Brusatte et al., 2014*). Nevertheless, we suggest that *Balaur* may have been proportionally shorter-tailed and with a less raptorial-looking foot than previously depicted (*Csiki et al., 2010*; *Brusatte et al., 2013*). Clearly, details of its cranial and dental anatomy are speculative. We assume that, like other paravians, *Balaur* was extensively feathered.

## CONCLUSIONS

The Maastrichtian paravian theropod *Balaur bondoc* is reinterpreted here as a basal avialan rather than as a dromaeosaurid. Features supporting its placement among Avialae include the hypertrophied and proximally placed coracoid tubercle, the anterior placement of the condyles of the humerus, the proximally fused carpometacarpus with a laterally shifted semilunate carpal, the closed intermetacarpal space, the reduced condyles on metacarpals I–II, the slender metacarpal III, the reduced phalangeal formula of the third digit, the extensively fused tibiotarsus, the extensively fused tarsometatarsus, the distal placement of the articular end of first metatarsal, the large size of the hallux, and the elongation of the penultimate phalanges of the pes. The absence of dromaeosaurid synapomorphies (e.g., non-ginglymoid metatarsals II and III, short metatarsal V) is thus interpreted as plesiomorphic, and not as the consequence of evolutionary reversal. Both its phylogenetic bracketing within basal avialans and the absence of predatory adaptations concur in indicating that *Balaur* was herbivorous or omnivorous, not predatory. The reduced forelimb of *Balaur* represents one of the most compelling pieces of evidence for insular adaptation in a Mesozoic bird. Furthermore, with its unique combination of features shared by distinct paravian clades and its possible placement as one of the closest relatives of Pygostylia, *Balaur* may represent a pivotal taxon in future investigations of Mesozoic bird interrelationships.

The hypothesis that some Mesozoic paravians represent the flightless descendants of volant, *Archaeopteryx*-like ancestors, most vigorously promoted by *Paul (1988)* and *Paul (2002)*, has not been supported by recent phylogenetic hypotheses (e.g., *Senter, 2007b*; *Turner, Makovicky & Norell, 2012*; *Agnolín & Novas, 2013*). Furthemore, phylogenetic analyses that incorporate sufficient character data are able to differentiate the members of such paravian lineages as Dromaeosauridae, Troodontidae and Avialae, as demonstrated by our present study. Nevertheless, reinterpretation of *Balaur* as a flightless avialan reinforces the point that at least some Mesozoic paravian taxa, highly similar in general form and appearance to dromaeosaurids, may indeed be the enlarged, terrestrialised descendants of smaller, flighted ancestors, and that the evolutionary transition involved may have required relatively little in the way of morphological or trophic transformation.

**Institutional Abbreviations**

ACUB    Museo di Anatomia Comparata, University of Bologna, Bologna, Italy
EME     Transylvanian Museum Society, Dept. of Natural Sciences, Cluj-Napoca, Romania

## ACKNOWLEDGEMENTS

We thank staff at the Transylvanian Museum Society (EME), Cluj-Napoca, in particular Matyas and Marta Vremir for allowing access to the *Balaur* holotype, for discussion and substantial invaluable assistance in Transylvania. DN and TB's work in Romania was funded by the National Geographic Society. Critical comments by James Clark, Michael Pittman, and Academic Editor John Hutchinson greatly improved the quality of the manuscript. We thank Steve Brusatte, Jonah Choiniere, Gareth Dyke and Corwin Sullivan for the detailed and critical comments on an earlier version of this manuscript. The program TNT is being made available with the sponsorship of the Willi Hennig Society. Jaime Headden kindly created and allowed use of the image in Fig. 7.

### Funding

The authors declare there was no funding for this work.

### Competing Interests

Andrea Cau is a volunteer associate researcher at the Museo Geologico e Paleontologico 'Giovanni Capellini, Bologna (Italy).

### Author Contributions

- Andrea Cau conceived and designed the experiments, performed the experiments, analyzed the data, contributed reagents/materials/analysis tools, wrote the paper, prepared figures and/or tables, reviewed drafts of the paper.
- Tom Brougham and Darren Naish analyzed the data, contributed reagents/materials/analysis tools, reviewed drafts of the paper.

### Supplemental Information

Supplemental information for this article can be found online at http://dx.doi.org/10.7717/peerj.1032#supplemental-information.

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
