# Peer review of "The phylogenetic affinities of the bizarre Late Cretaceous Romanian theropod Balaur bondoc (Dinosauria, Maniraptora): dromaeosaurid or flightless bird?"

_PeerJ, doi:10.7717/peerj.1032_

## Round 0.1 · original submission · Major Revisions

We've obtained two non-anonymous reviews with quite helpful constructive criticisms of the paper, and they are in agreement that the paper should eventually be published. However, major revisions are needed. In particular, I agree with reviewer 1 that the latest dataset by Brusatte et al. (2014) must be analyzed in the revised paper. In a fast-moving area of research like this, and considering the huge uncertainties in relationships around the base of Paraves/Avialae, any paper whose fundamental point is that 1 taxon's position in the phylogeny must use the latest data and be circumspect in its treatment of other recent datasets/analyses. The other suggestions for changes all seem very reasonable and so they should be in the revised paper. Please address all comments by the reviewers in a point-by-point Response document. The paper will need to be re-reviewed hopefully by the same reviewers, and hopefully only briefly, but that all depends on the assiduousness of the revision. We look forward to seeing that revised manuscript.

·

Basic reporting

The authors report their work to high international standards:

-The article is written in English using clear and unambiguous text and conforms to professional standards of courtesy and expression.

-The article include substantial introduction and background to demonstrate how the work fits into the broader field of knowledge. Extensive prior literature is appropriately referenced. For the aforementioned points to be sufficiently met - see comment on line 72 of the reviewer PDF - two important papers relating to the study need to be more extensively treated/discussed in the manuscript: Brusatte et al. (2014) and Xu et al. (2015).

Some figures needed additional labeling, but this is a minor issue.

The submission is ‘self-contained,’ and represents an appropriate ‘unit of publication’, and includes most of the results relevant to the hypothesis (but see comment on line 72 of the reviewer PDF).

Experimental design

The submission describes original primary research within the scope of the journal.

The submission clearly defines the research question, and is relevant and meaningful.

The investigation was conducted rigorously and to a high technical standard, despite the use of an older dataset (see comment on line 72 of the reviewer PDF).

The authors should be commended on their inclusion of detailed methodological information in the manuscript and supporting information.

Validity of the findings

The findings of the paper are well-argued and well-supported by thorough analyses of the datasets used. However, as mentioned in the comment on line 72 of the reviewer PDF, the latest dataset by Brusatte et al. (2014) is a major update on the Turner et al. (2012) that was used and its analysis revealed important changes to the interrelationships between and amongst dromaeosaurids, troodontids and avialans, and our confidence in them. As such, despite the high quality of the submitted work, the validity of the manuscript's main finding - that Balaur is a avialan - cannot be confirmed by the TWiG dataset until the existing analysis is replaced by one using a modified Brusatte et al. (2014) dataset (the latest TWiG dataset).

Additional comments

An analysis using a modified Brusatte et al. (2014) dataset will be very time consuming so I recommend reformatting the manuscript using support from the modified Lee et al. (2014) analysis only since this is already a valuable contribution. Then, the authors can perhaps follow-up with a second paper showing the results of the analysis of the modified Brusatte et al. (2014) dataset and commenting on how these differ from those of the previous paper.

·

Basic reporting

The authors of this paper would have us believe that Mark Norell and Steve Brusatte, two renowned experts on theropod dinosaurs closely related to birds, misinterpreted the phylogenetic relationships of the Romanian theropod Balaur and did not recognize that it was a bird rather than a dromaeosaurid. After reading the paper I believe them.

The paper is well written and the relevant characters are discussed at a length appropriate to demonstrating their distribution. I am not an expert on basal birds so hopefully someone familiar with their anatomy will review this. I’ve made several notes on the ms. and supplementary information and will only discuss a few issues here.

1) Throughout much of the paper the authors refer to “derived taxa”. Characters are derived, taxa are not. Elsewhere in the paper they instead mention “crownward taxa,” which is more precise and doesn’t assume linear evolution towards a particular morphology.

2) The paper states (line 100):
“In order to test whether assumptions on character weighting influence the placement of Balaur among Paraves, both datasets were subjected to implied weighting analyses.”
However, their only assumption about character weighting was equal weighting, which isn’t much of an assumption. They might mean they are testing whether the assumption of equal weights influences its placement, but what they are really doing is downweighting homoplasious characters, a specific kind of character weighting. They should better explain what it is they think this is testing.

3) For the character (line 158) “Humerus longer than half the combined length of tibiotarsus and tarsometatarsus” it isn’t clear to me why this ratio is biologically meaningful, and whether it indicates a long humerus or a short lower leg. If it is a measure of relative humerus length or lower leg length then why not compare it to something not so directly involved in locomotion (e.g., vertebral length)?

4) Lines 286-289 – Manual ungual I is shown to lack a particular aspect of dorsal curvature when the articular surface is held vertical, but it does not appear to be any less curved than the microraptorine claw illustrated in figure S1. Describing it as exhibiting a “reduction” in curvature is not accurate, rotation of the articular surface would result in the derived character without any change in curvature.

5) Lines 642-644 - “The definitions of many characters used in the analysis of Turner et al. (2012) impose congruence by linking more than one variable character to a particular state (see Brazeau 2011 and references therein), or by mixing together neomorphic and transformational characters as alternative states of the same character statements.” This needs some examples to demonstrate the flaws.

6) The specimen illustrations are all somewhat crude, but some of this is due to the poor preservation of the specimen and its irregular exterior surface. They suffice for the purposes of the paper but could be improved with better shading.

7) The skull on the new reconstruction by Jaime Headden looks like a dromaeosaur rather than Archaeopteryx or Sapeornis.

Experimental design

See above

Validity of the findings

See above

---

## Round 0.2 · Minor Revisions

The reviewers have made constructive final comments and I think this would become an acceptable paper if their final recommendations are dealt with well. With reviewer 1, I agree, just cite the new Xu paper but no need to take it fully into account. Likewise, reviewer 2's recommendation on the ratio character is a good one. It's a dodgy character but you could still use it if you justify it. Maybe your results are not sensitive to whether it is used or not? That's an important point.

Please edit accordingly and send us a point-by-point Response to all Reviewers' comments. If I am convinced, I will recommend that the study is accepted and there thus would be no need for further review.

·

Basic reporting

The authors report their work to high international standards. The article is written in English using clear and unambiguous text and conforms to professional standards of courtesy and expression. The article includes substantial introduction and background to demonstrate how the work fits into the broader field of knowledge. Extensive prior literature is appropriately referenced.

However, Xu et al.'s (2015) recent paper on the taxonomic status of Linheraptor still needs to been referenced, even if the author's disagree with this paper's conclusions. This is because if Linheraptor is a valid taxon - as Xu et al. (2015) argue - then the codings for Tsaagan in the Brusatte et al. (2014) dataset [and Turner et al. (2012)'s] represent a chimeric taxon of Linheraptor + Tsaagan that potentially produced spurious results. Far for requesting for this taxonomic issue to be discussed in detail in this study (which would deviate from the main focus of the paper) I simply suggest an acknowledgement of Xu et al. (2015)'s results. For example, in the Supplementary Information where the Brusatte et al. (2014) matrix is being discussed: 'Xu et al. (2015) recently provided further support for Linheraptor as a valid dromaeosaurid taxon which Brusatte et al. (2014) consider to be a junior synonym of Tsaagan [Brusatte et al. (2014) adopted the same taxonomic opinion of Turner et al. (2012) as they used Turner et al.'s (2012) codings of Tsaagan]. As we have not studied this specimen in person we have adopted Brusatte et al.'s (2014) taxonomic opinion on Linheraptor pending a detailed phylogenetic analysis with Linheraptor and Tsaagan as separate taxa.' I (Michael Pittman) am the second author of the Xu et al. (2015) paper but I suggest this change not from a selfish interest in gaining additional citations, but from our duty as taxonomists to convey all relevant published taxonomic opinion so that taxonomic issues can be resolved using all of the data available.

Experimental design

The submission describes original primary research within the scope of the journal.

The submission clearly defines the research question, and is relevant and meaningful.

The investigation was conducted rigorously and to a high technical standard.

The authors should be commended on their inclusion of detailed methodological information in the manuscript and supporting information.

Validity of the findings

The findings of the paper are well-argued and well-supported by thorough analyses of the datasets used. I am very happy with the revisions the authors have made.

Additional comments

If the suggested change in the 'Basic Reporting' section is made as well as the following two points I strongly support acceptance of this article for publication:

The two points below are aimed at giving more background information about the new and revised characters used. This information will be invaluable for users of the revised character list, particularly students who might not be as familiar with the anatomy being discussed.

1. Character state 4 of Character 147 (splint metacarpal bearing no phalanges) presumably refers to the condition in the parvicursorine theropod Linhenykus. As this theropod has a very usual hand whose anatomy might be unfamiliar to people using the matrix, I recommend adding the Xu et al. (2013) citation as this contains the clearest images of the anatomy discussed (http://www.app.pan.pl/article/item/app20110083.html).

2. Characters 436 and 859 both mention relative length in terms of a percentage. I think it would be really useful to briefly explain how and why you chose the cut off that you used e.g. are there currently two distinct rather than continuous clusters of percentage values about his cut-off; are the states mentioned peppered across the theropod tree or are they important for characterising particular clades.

I would like to convey - once again - that I am very happy with the edits that the authors have made. With the three changes requested in this re-review, this important paper should be swiftly published.

·

Basic reporting

no comments

Experimental design

no comment

Validity of the findings

no comment

Additional comments

The revisions are mostly fine, although I am still perplexed by character 158 (ratio of humerus to tibiotarsus+tarsometarsus length). Their argument is that this character (using femoral length rather than distal hindlimb length) is used in other analyses, but they modify other characters they perceive as mistaken so they must accept it as a valid character. They modify this character to apply to the distal limb rather than the femur because the femur isn't preserved in Balaur, but it results in what seems to be a random ratio, especially without any documentation that distal hindlimb length is correlated with femoral length. They need to justify this character better.

---

## Round 0.3 · accepted · Accept

The authors have done a fine job of revising the MS following the reviews. Where they deviate from the recommendations, their justifications seem sound to me. So I am comfortable to accept the MS now - congratulations!